# Adipocyte HSL is required for maintaining circulating vitamin A and RBP4 levels during fasting

Julia S Steinhoff[1,9], Carina Wagner[2,9], Henriette E Dähnhardt[1], Kristina Košić[2], Yueming Meng[1], Ulrike Taschler [2], Laura Pajed[2], Na Yang [1], Sascha Wulff[1], Marie F Kiefer[1], Konstantin M Petricek [1], Roberto E Flores[1], Chen Li[1], Sarah Dittrich [1], Manuela Sommerfeld[1], Hervé Guillou [3], Andrea Henze[4,5], Jens Raila [6], Sylvia J Wowro [1], Gabriele Schoiswohl [7], Achim Lass [2,8✉] & Michael Schupp [1✉]

## Abstract

**Vitamin A (retinol) is distributed via the blood bound to its specific carrier protein, retinol-binding protein 4 (RBP4). Retinol-loaded RBP4 is secreted into the circulation exclusively from hepatocytes, thereby mobilizing hepatic retinoid stores that represent the major vitamin A reserves in the body. The relevance of extrahepatic retinoid stores for circulating retinol and RBP4 levels that are usually kept within narrow physiological limits is unknown. Here, we show that fasting affects retinoid mobilization in a tissue-specific manner, and that hormone-sensitive lipase (HSL) in adipose tissue is required to maintain serum concentrations of retinol and RBP4 during fasting in mice. We found that extracellular retinol-free apo-RBP4 induces retinol release by adipocytes in an HSL-dependent manner. Consistently, global or adipocyte-specific HSL deficiency leads to an accumulation of retinoids in adipose tissue and a drop of serum retinol and RBP4 during fasting, which affects retinoid-responsive gene expression in eye and kidney and lowers renal retinoid content. These findings establish a novel crosstalk between liver and adipose tissue retinoid stores for the maintenance of systemic vitamin A homeostasis during fasting.**

**Keywords** Fasting; HSL; RBP4; Retinol; Vitamin A
**Subject Category** Metabolism

## Introduction

Vitamin A comprises retinol and its biologically active metabolites (retinoids) with essential functions for human life (McCollum and Davis, 1913). In the eye, 11-*cis* retinaldehyde is the light-sensitive chromophore of the rhodopsin complex and required for the visual cycle in the retina (Palczewski et al, 2000). All-*trans* retinoic acid (atRA) and its derivatives regulate reproduction, embryonic development, immune cell function, and cell proliferation and differentiation (Blaner, 2007; Blaner et al, 1999; Morriss-Kay and Sokolova, 1996; Napoli, 1996) as activating ligands for nuclear retinoic acid and retinoid X receptors (RAR and RXR) (Heyman et al, 1992; Levin et al, 1992; Petkovich et al, 1987; Rühl et al, 2015).

All vitamin A in the body derives from nutritional ingestion of carotenoid precursors or retinoids, in case of the latter transported and delivered via lipoproteins/chylomicrons and the action of membrane-bound lipoprotein lipase (Quadro et al, 2004; von Lintig et al, 2020). Vitamin A stores are formed by the esterification of retinol with fatty acids and its storage in organs like liver and adipose tissue, with liver-resident stellate cells containing the majority of retinyl ester stores of the body. The mobilization of retinyl esters requires their hydrolytic cleavage and several lipases have been implicated in the hydrolysis of retinyl esters (Grumet et al, 2016; Schreiber et al, 2012). The rate-limiting enzyme in liver is unknown, however, hormone-sensitive lipase (HSL) is the major retinyl ester hydrolase (REH) in adipose tissue (Strom et al, 2009).

Hepatic mobilization of retinol and its transport in the circulation is facilitated by retinol-binding protein 4 (RBP4, also known as RBP), the specific transport protein of retinol in the blood (Kanai et al, 1968). Upon loading of RBP4 with retinol within the hepatocyte, a complex is formed with transthyretin (TTR) prior secretion (Bellovino et al, 1996; Monaco et al, 1995; Naylor and Newcomer, 1999). In the circulation, retinol-loaded holo-RBP4 binds its membrane receptors "stimulated by retinoic acid 6" (STRA6) and STRA6-like (STRA6L, also referred to as RBP receptor 2) for delivery of retinol to target cells (Alapatt et al, 2013; Kawaguchi et al, 2011). Moreover, STRA6 also mediates retinol efflux to apo-RBP4 and retinol exchange to replenish cellular retinoids (Kawaguchi et al, 2012), underlining the dynamics of retinol transport and distribution to warrant retinoid homeostasis.

[1]Charité Universitätsmedizin Berlin, corporate member of Freie Universität Berlin and Humboldt-Universität zu Berlin, Institute of Pharmacology, Max Rubner Center (MRC) for Cardiovascular-Metabolic-Renal Research, Berlin, Germany. [2]Institute of Molecular Biosciences, NAWI Graz, University of Graz, Graz, Austria. [3]Toxalim (Research Center in Food Toxicology), INRAE, ENVT, INP- PURPAN, UMR 1331, UPS, Université de Toulouse, Toulouse, France. [4]Martin Luther University Halle-Wittenberg, Institute of Agricultural and Nutritional Sciences, Halle, Germany. [5]Junior Research Group ProAID, Institute of Nutritional Science, University of Potsdam, Nuthetal, Germany. [6]Department of Physiology and Pathophysiology, Institute of Nutritional Science, University of Potsdam, Nuthetal, Germany. [7]Gottfried Schatz Research Center, Molecular Biology and Biochemistry, Medical University of Graz, Graz, Austria. [8]BioTechMed-Graz, Graz, Austria. [9]These authors contributed equally: Julia S Steinhoff, Carina Wagner. ✉E-mail: achim.lass@uni-graz.at; michael.schupp@charite.de

Mice that lack RBP4 accumulate retinol and retinyl esters in liver and have low levels of retinol in the circulation. Retinal function and visual acuity are impaired during early life in these mice, normalizing after 4–5 months when fed a vitamin A-sufficient diet (Quadro et al, 1999). In fact, although expressed in many tissues and cell types, RBP4 in the circulation is entirely hepatocyte-derived, since it is undetectable in the serum of mice with hepatocyte-specific deletion of the gene (Thompson et al, 2017). Accordingly, when overexpressed in hepatocytes, RBP4 is readily secreted and increases circulating retinol levels while depleting hepatic retinyl esters (Muenzner et al, 2013; Steinhoff et al, 2022b).

In C57BL/6 mice, serum concentrations of retinol are kept constant at ~0.5–1.5 μM, even in stress situations such as prolonged nutritional vitamin A deficiency that deplete hepatic retinyl ester stores before affecting serum retinol concentrations (Gao et al, 2019; Liu and Gudas, 2005). The regulation of hepatic retinol mobilization and how it coordinates with peripheral retinoid stores such as those in adipose tissue is unexplored.

In this study, we asked whether prolonged fasting affects retinol mobilization from liver and hypothesized that the absence of vitamin A ingestion would be compensated by increased hepatic mobilization. To our surprise, we found that 24 h of fasting in mice did not deplete hepatic retinyl ester stores, but resulted in reduced REH activity accompanied by an accumulation of RBP4 protein in liver, suggesting lower hepatic retinol mobilization. In contrast, retinoid content of adipose tissue declined, maintaining serum retinol and RBP4 at steady-state levels. Mobilization of adipose tissue retinoids was dependent on HSL, because mice with global or adipocyte-specific deletion of HSL failed to maintain serum retinol and RBP4 levels during fasting. Lower retinol and RBP4 in the circulation associated with altered expression of atRA-responsive genes in the eye and kidney and reduced renal retinoid content. These findings reveal an unappreciated crosstalk between liver and adipose tissue retinoid stores, vastly extending our understanding of retinoid homeostasis and HSL-dependent pathologies.

## Results

### Fasting induces RBP4 and lowers TTR in liver

In order to investigate the effects of nutrient deprivation on hepatic retinol mobilization, male C57BL/6 J mice were fed ad libitum or fasted for 24 h starting at Zeitgeber time (ZT) 8 (Fig. 1A). Fasted mice showed the expected drop in blood glucose and elevation in serum NEFA levels (Fig. 1B), validating the experimental setup. We found that serum retinol was not affected by 24 h of fasting (Fig. 1C). Expression of phosphoenolpyruvate carboxykinase 1 (Pck1), rate limiting for hepatic gluconeogenesis, displayed the expected upregulation. Interestingly, mRNA expression of Rbp4 was induced whereas that of Ttr was downregulated (Fig. 1D). Hepatic REH activity, providing retinol for RBP4 loading and subsequent complex formation with TTR, was analyzed ex vivo using liver lysates (1000 × g supernatant) and showed a decrease in fasted mice (Fig. 1E). We next determined abundance of RBP4 and TTR protein in liver and detected a robust increase in RBP4, exceeding its increase in mRNA, and a concomitant reduction in TTR (Fig. 1F,G). For correct analysis, protein extracts needed to be boiled for 20 min to fully denature notoriously stable TTR

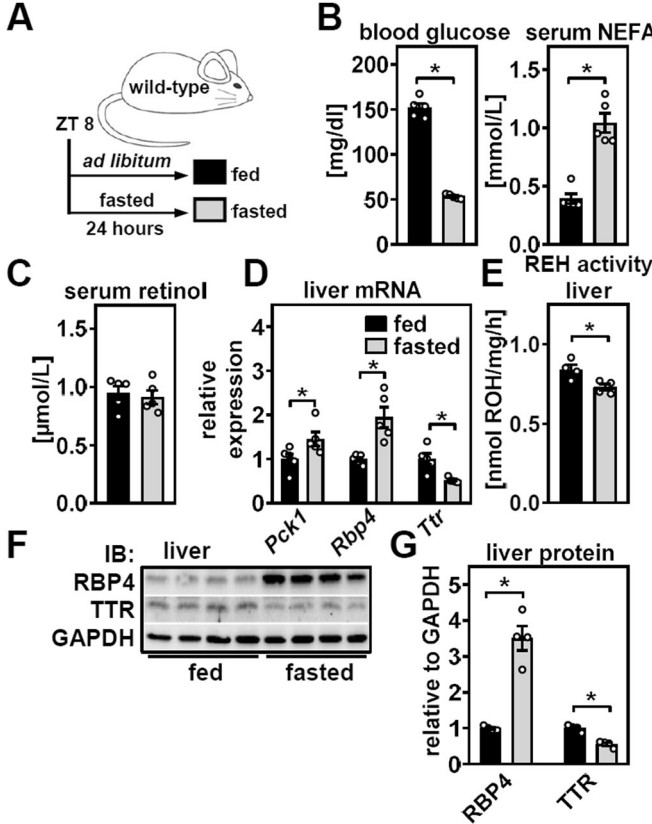

Figure 1. Fasting induces RBP4 and lowers TTR in the liver.

(A) Male C57BL/6J mice were fed ad libitum or fasted for 24 h as depicted. (B) Blood glucose and non-esterified fatty acids (NEFA) in fed and fasted mice were determined. (C) Serum retinol concentrations were analyzed by HPLC. (D) Hepatic expression of indicated genes was measured by qPCR. (E) Retinyl ester hydrolase (REH) activity of liver lysates from fed and fasted mice was analyzed ex vivo. (F) Protein abundance of RBP4 and TTR in livers of fed and fasted mice, GAPDH served as loading control. (G) The densitometric analysis of proteins shown in (F). Data information: Data are represented as individual data points of n = 5, 5 (B–D) or n = 4, 4 (E–G) biological replicates and mean ± s.e.m. and *P < 0.05 vs. ad libitum-fed mice using an unpaired two-tailed Student's t test. Source data are available online for this figure.

complexes, and serum TTR was used to identify the corresponding protein band in liver extracts (Appendix Fig. S1). Increased hepatic RBP4 protein was not irreversible and due to liver damage upon prolonged fasting since subsequent refeeding for 18 h completely reversed this effect (Fig. EV1A–D). Thus, while serum retinol levels remain constant, hepatic RBP4 increases whereas TTR is down-regulated after prolonged fasting. These findings, in the context of the observed reduction in REH activity, point towards an overall lower hepatic retinol mobilization and RBP4 secretion upon fasting, leading to an accumulation of RBP4 protein in the liver.

### Fasting does not affect total RBP4 but reduces TTR and slightly increases apo-RBP4 in the circulation

Consistent with unchanged serum retinol, circulating levels of RBP4 remained unaltered in fasted mice (Fig. 2A,C). Serum TTR was reduced (Fig. 2A,C), as reported previously for fasted rats (Wade et al, 1988). Detergent-free native immunoblotting of serum

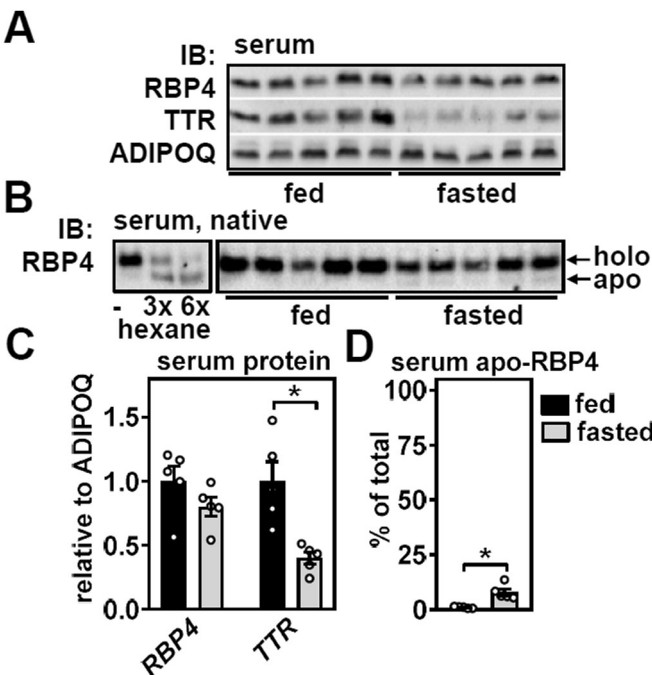

**Figure 2. Fasting does not affect total RBP4 but reduces TTR and increases apo-RBP4 in the circulation.**

(A) Serum RBP4 and TTR levels of fed and fasted mice were analyzed by immunoblotting, and serum ADIPOQ served as loading control. (B) Abundance of apo- and holo-RBP4 in serum of fed and fasted mice was analyzed by SDS-free native immunoblotting (right panel). Repeated hexane-extractions of serum was used as a control for visualization of each isoform (left panel). (C) Densitometric analysis of blots shown in (A). (D) Densitometric analysis of blots shown in (B). Data information: Data are represented as individual data points of $n = 5, 5$ (A–D) biological replicates and mean ± s.e.m. and *$P < 0.05$ vs. ad libitum-fed mice using an unpaired two-tailed Student's $t$ test. Source data are available online for this figure.

can differentiate between holo- and apo-RBP4, which we validated by depleting serum retinol by repeated hexane extractions that result in the expected increase in apo-RBP4 (Fig. 2B, left panel). Consistent with earlier observations (Muenzner et al, 2013), serum RBP4 circulates almost exclusively as retinol-containing holo-RBP4 in ad libitum-fed mice (Fig. 2B, right panel and D). However, we detected a slight increase in the faint but discernible band corresponding to apo-RBP4 in fasted mice, suggesting that retinol saturation of serum RBP4 is slightly reduced (Fig. 2B, right panel and D). We conclude that fasting has minor effects on circulating RBP4, albeit slightly increasing retinol-free apo-RBP4. Serum TTR was reduced by more than 50%, most likely due to reduced hepatic expression and secretion.

## Expression and secretion of RBP4 in primary hepatocytes is regulated by cAMP and FOXO1

We took advantage of primary mouse hepatocytes to dissect the fasting-dependent regulation of RBP4 expression and secretion. Pharmacological activation of peroxisome proliferator-activated receptor α (PPARα), a nuclear receptor known to translate hepatic NEFA influx derived from adipose tissue into hepatic gene expression during fasting (Fougerat et al, 2022; Kersten et al,

1999; Regnier et al, 2018), induced mRNA expression of the canonical PPAR-target and fasting-induced gene Retinol Saturase (*RetSat*) (Weber et al, 2020), but not that of *Rbp4* (Fig. 3A). Consistent with a PPARα-independent mechanism, liver *Rbp4* mRNA expression remains fasting inducible in mice that lacked PPARα in hepatocytes (Smati et al, 2020). Also atRA, increasing the expression of the RAR target "cytochrome P450 family 26 subfamily A member 1" (*Cyp26a1*) in a concentration-dependent manner, had no effect on *Rbp4* expression (Fig. 3B) or its secretion in primary hepatocytes (Fig. EV2A,B). In conjunction with PPARα, activation of glucocorticoid receptor via increased corticosterone in mice (Schupp et al, 2013) and increased cAMP signaling (Altarejos and Montminy, 2011) shape the hepatic response to fasting. Combined incubation with the synthetic glucocorticoid dexamethasone (Dex) and the stable cAMP analogue 8-Br-cAMP potently induced *Pck1*, but also increased expression of *Rbp4* (Fig. 3C). Of both pathways, exposure to the cAMP analogue was sufficient to achieve full inducibility of *Rbp4*, but not of *Pck1* mRNA, suggesting that cAMP signaling alone mediates the upregulation of *Rbp4* (Fig. 3C). Notably, basal *Rbp4* expression in primary hepatocytes remains very high when compared to liver and also compared to *Pck1*, whose transcript levels drop significantly after hepatocyte isolation and accounting for the high inducibility of *Pck1* in vitro by hormonal stimulation due to its low basal expression. Hepatocyte cAMP is regulated downstream of glucagon receptor signaling and, indeed, glucagon exposure induced both *Pck1* and *Rbp4* mRNA expression (Fig. 3D). These findings are in accordance with earlier reports, demonstrating cAMP-dependent upregulation of *Rbp4* in murine hepatoma cells and elevated *Rbp4* expression in mouse liver upon glucagon injection (Bianconcini et al, 2009; Jessen and Satre, 1998). Whether cAMP affects RBP4 secretion by hepatocytes has not been reported and was investigated next. Surprisingly, despite the fact that 8-Br-cAMP induced *Rbp4* mRNA expression, cellular RBP4 protein content was unaltered while it was dramatically decreased in conditioned media of hepatocytes (Fig. 3E,F). An accumulation of cellular RBP4 protein upon 8-Br-cAMP treatment, however, was observed in the presence of a proteasome inhibitor (Fig. EV2C,D), indicating that higher amounts of RBP4 protein in primary hepatocytes are targeted by proteasomal degradation. We conclude that the fasting-associated increase in the liver *Rbp4* mRNA expression is likely due to glucagon/cAMP signaling, which also potently reduces its secretion.

Part of cAMP's hepatic action is mediated by the forkhead transcription factor FOXO1 (Matsumoto et al, 2007), which was previously shown to regulate genes involved in retinoid homeostasis in liver (Obrochta et al, 2015). We depleted Foxo1 by two different siRNA oligonucleotides in primary hepatocytes, reducing its mRNA and protein expression by >75% (Fig. 3G,H). This reduction was accompanied by a robust decrease of its known target gene "insulin like growth factor binding protein 1" (*Igfbp1*) (Durham et al, 1999). Likewise, mRNA expression of *Rbp4* was reduced (Fig. 3H), identifying *Rbp4* as novel FOXO1 target in hepatocytes. Consistently, inactivating FOXO1 by insulin (Nakae et al, 2000) lowered expression of both *Igfbp1* and *Rbp4* in primary hepatocytes (Fig. 3I). Regulation of *Rbp4* expression by FOXO1 was previously reported in a β-cell line (Miyazaki et al, 2012). Interestingly, hepatocyte-specific deletion of the insulin receptor in mouse liver, thereby preventing FOXO1 inactivation, rendered

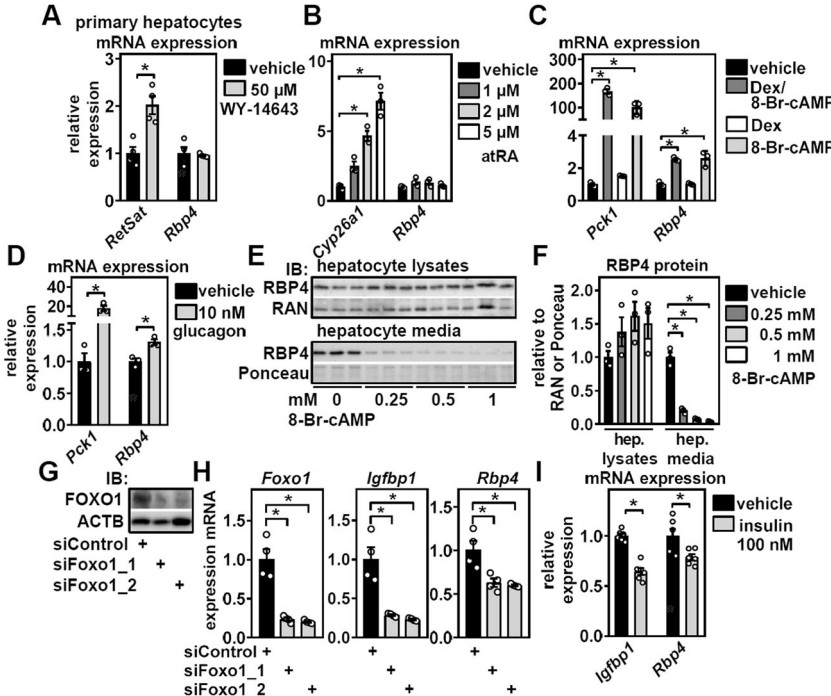

**Figure 3. Expression and secretion of RBP4 in primary hepatocytes is regulated by cAMP and FOXO1.**

Primary hepatocytes were isolated from mouse liver and treated with vehicle or (**A**) the synthetic PPARα agonist WY14643, (**B**) all-*trans* retinoic acid (atRA), (**C**) dexamethasone (Dex), 8-Br-cAMP, and (**D**) glucagon as indicated for 24 h. mRNA expression of appropriate control genes and *Rbp4* were determined by qPCR. (**E**) Hepatocytes were incubated with increasing concentrations of 8-Br-cAMP and RBP4 protein in cell lysates and media analyzed by immunoblotting, RAN protein and Ponceau membrane staining served as loading controls, respectively. (**F**) Densitometric analysis of blots shown in (**E**). (**G**) Primary hepatocytes were treated with non-targeting (Control) or Foxo1 siRNA and FOXO1 protein analyzed by immunoblotting, ACTB served as loading control. (**H**) mRNA expression of indicated genes in hepatocytes described in (**G**) was analyzed by qPCR. (**I**) Primary hepatocytes were treated with insulin as indicated for 24 h and mRNA expression of *Igfbp1* and *Rbp4* determined by qPCR. Data information: In (**A–H**) Data are representative for three independent experiments and performed in triplicates (**A–F**), or quadruplicates (**H**). (**I**) Data are representative of two independent experiments performed with six replicates. Data are represented as individual data points and mean ± s.e.m. and *$P < 0.05$ vs. vehicle- or siControl-treated hepatocytes using an unpaired two-tailed Student's *t* test (**A, D, I**) or one-way ANOVA with Dunnett's correction for multiple testing (**B, C, F, H**). Source data are available online for this figure.

*Rbp4* mRNA expression constitutively high and unresponsive to fasting in male mice (Smati et al, 2020), corroborating the regulation of *Rbp4* by FOXO1 and its relevance for the fasting response.

## Fasting reduces retinoid levels in white adipose tissue but not in the liver or lung

We next quantitated retinol and major retinyl esters in the liver. To our surprise, hepatic retinyl ester concentrations were significantly increased in mice that were fasted for 24 h (Fig. 4A, left panel). Taken into account that prolonged fasting, besides reducing body weights, has distinctive effects on organ/tissue weights (Appendix Fig. S2), we multiplied retinoid concentrations with total tissue mass. After mass correction, there was no significant change in hepatic retinol or retinyl ester content in fasted mice (Fig. 4A, right panel). Also in the lung, another organ with substantial retinoid storage (Shmarakov et al, 2023), fasting had no effect on total retinoid abundance (Fig. 4B). In contrast, fasting led to decreased retinol and retinyl ester content in both inguinal and perigonadal white adipose tissue (ing/pgWAT), with comparable tissue concentrations but strongly reduced total WAT mass in fasted mice (Fig. 4C,D; Appendix Fig. S2). Thus, 24 h of fasting depletes retinoids in WAT but not those stored in the liver or lung.

## HSL is required for mobilization of WAT retinoids and to maintain circulating RBP4 and retinol concentrations during fasting

To mobilize retinol into the circulation, stored tissue retinyl esters require prior hydrolysis. We therefore analyzed REH activities in tissue lysates of 16 h-fasted mice in the presence of pharmacological inhibitors of adipose tissue triglyceride lipase (ATGL) and HSL, major acylglycerol lipases that also exhibit REH activity (Taschler et al, 2015; Wei et al, 1997). Inhibition of ATGL by Atglistatin (iATGL) (Mayer et al, 2013) had no effect on hepatic and adipose REH activities, suggesting that ATGL is dispensable for retinol mobilization from liver and WAT (Fig. 5A). In contrast, inhibition of HSL by NNC-0076-0000-0079 (iHSL) (Schweiger et al, 2006) slightly decreased REH activity in liver lysates but almost completely abolished REH activity in WAT (Fig. 5A). We conclude that HSL provides the quantitatively most important REH activity in WAT but not in the liver. Moreover, fasted mice with germline deletion of *Hsl*, lacking HSL expression in all tissues, including liver and pgWAT (Fig. EV3A) (Haemmerle et al, 2002) showed a modest increase in liver REH activity but almost undetectable REH activity in WAT (Fig. 5B), which is consistent with previous findings (Strom et al, 2009). In accordance with defective retinyl ester

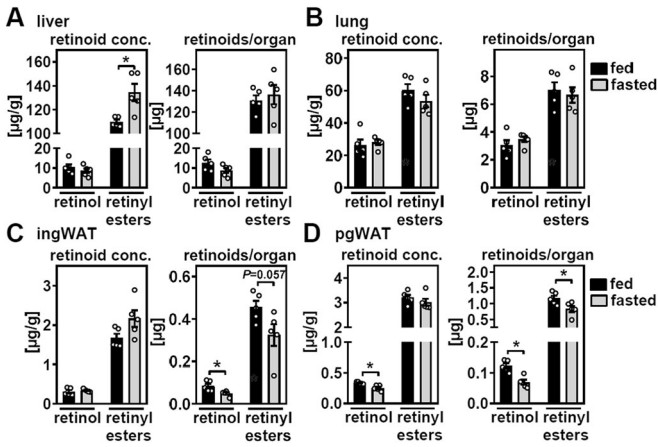

**Figure 4. Fasting reduces retinoid levels in white adipose tissue but not in liver or lung.**

Mice were fed ad libitum or fasted for 24 h and tissue retinol and retinyl esters in (A) liver, (B) lung, (C) inguinal-, and (D) perigonadal white adipose tissue (ing/pgWAT) analyzed by HPLC. Retinoids are normalized to tissue weight (left panel) or shown as µg per total organ (right panel). Data information: Data are represented as individual data points of $n = 5$, 5 biological replicates and mean ± s.e.m. and *$P < 0.05$ vs. ad libitum-fed mice using an unpaired two-tailed Student's $t$ test. Source data are available online for this figure.

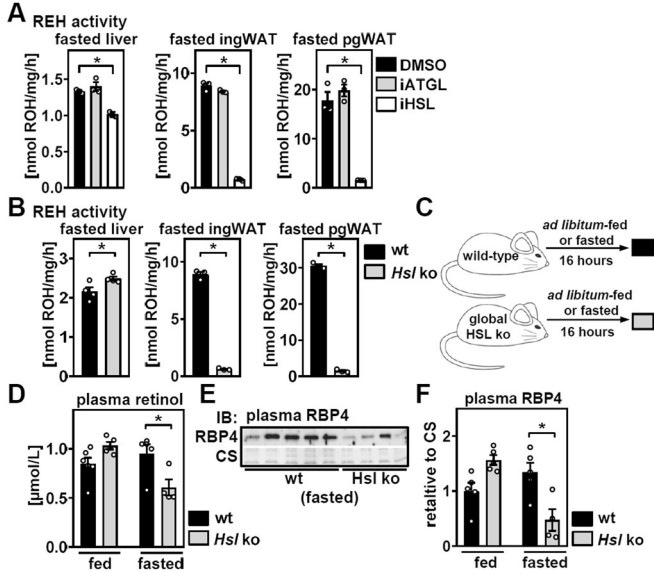

**Figure 5. HSL is required for mobilization of WAT retinoids and to maintain circulating RBP4 and retinol concentrations during fasting.**

(A) Retinyl ester hydrolase (REH) activity of liver and white adipose tissue (WAT) lysates of 16 h-fasted wild-type mice (wt) in the presence of DMSO vehicle, the ATGL inhibitor Atglistatin, or the HSL inhibitor NNC-0076-0000-0079 was analyzed ex vivo. Wild-type (wt) mice or mice with global *Hsl* deletion were fasted for 16 h and (B) retinyl ester hydrolase (REH) activity of indicated tissues analyzed ex vivo. (C) Wt mice or mice with global *Hsl* deletion were fasted as indicated and (D) plasma retinol analyzed by HPLC. (E) Plasma RBP4 was determined by immunoblotting. Coomassie staining (CS) served as loading control. (F) Densitometric analysis of plasma RBP4 in ad libitum-fed and 16 h-fasted mice. Data information: Data are representative for three independent experiments, performed in triplicates (A). (B) Data show biological replicates with $n = 4$, 4 (liver) and $n = 3$, 3 (ing and pgWAT), in (D), data show biological replicates with $n = 6$, 5, 5, 4, and in (F), data show biological replicates with $n = 5$, 5, 5, 4. Data are presented as mean ± s.e.m. and *$P < 0.05$ vs. DMSO vehicle or wt mice using a one-way ANOVA with Dunnett's correction for multiple testing (A), an unpaired two-tailed Student's $t$ test (B), or a two-way ANOVA with Sidak's correction for multiple testing (D, F). Source data are available online for this figure.

hydrolysis in WAT but not the liver, HSL-deficient mice had lower levels of retinol and a robust accumulation of retinyl esters in pgWAT but not liver (Fig. EV3B), and as reported previously (Strom et al, 2009).

Upon fasting, HSL activity is induced by catecholamines and other lipolytic hormones via cAMP-dependent protein kinase A-dependent phosphorylation (Clifford et al, 2000; McKnight et al, 1998; Sztalryd and Kraemer, 1994). We therefore analyzed circulating retinol and RBP4 levels in ad libitum-fed (= low HSL activity) and 16 h-fasted (= high HSL activity) mice (Fig. 5C). Loss of HSL did not affect circulating retinol concentrations or RBP4 levels in ad libitum-fed animals (Fig. 5D–F, RBP4 blots for ad libitum-fed mice shown in Appendix Fig. S3A). However, fasted mice lacking HSL failed to maintain their retinol and RBP4 levels in the circulation (Fig. 5D–F), suggesting that HSL is required for retinyl ester hydrolysis and retinoid homeostasis specifically during fasting. In liver, deletion of HSL trended to reduce RBP4 protein abundance during fasting (Fig. EV3C–E). In contrast, mRNA and protein expression of RBP4 in pgWAT was not regulated by fasting but reduced in mice lacking HSL (Fig. EV3F,H,I), potentially as a consequence of impaired PPAR activation (Rosell et al, 2012) upon the loss of HSL that was reported previously (Pajed et al, 2021; Strom et al, 2009) and evidenced by lower expression of canonical PPAR target genes (Fig. EV3G).

To test whether livers of HSL knockout mice were per se defective in mobilizing retinyl ester stores, we fed wt mice and mice globally lacking HSL a vitamin A-deficient diet (VAD) (Fig. EV4A). After 9 weeks of VAD, plasma retinol concentrations in ad libitum-fed mice remained within the physiologic range and were not affected by global deletion of HSL (Fig. EV4B). On the other hand, overnight fasting reduced plasma retinol concentrations in mice lacking HSL (Fig. EV4C), as observed in fasted mice kept on normal chow (Fig. 5D). In the liver, retinol and retinyl ester contents were depleted

by feeding VAD to the same extent in both genotypes (Fig. EV4D). This suggests that hepatic mobilization of retinyl esters upon feeding VAD is not compromised in mice that lack HSL. Taken together, we conclude that the global loss of HSL lowers circulating RBP4 and retinol concentrations during fasting, presumably via impaired retinyl ester hydrolysis and retinol mobilization from WAT.

## Retinol-free apo-RBP4 induces HSL-dependent retinol release from adipocytes

The exact mechanism by which adipocytes release retinol is unknown (Steinhoff et al, 2022a; Wei et al, 1997). Serum RBP4 is liver-derived (Thompson et al, 2017), suggesting that WAT, in contrast to liver, does not utilize holo-RBP4 secretion for retinol release. Alternatively, circulating retinol-free apo-RBP4, derived from holo-RBP4 upon retinol delivery, may function as a high-affinity acceptor for cellular retinol molecules, as demonstrated previously for STRA6-mediated loading of apo-RBP4 (Isken et al, 2008; Kawaguchi et al, 2012; Muenzner et al, 2013). We therefore

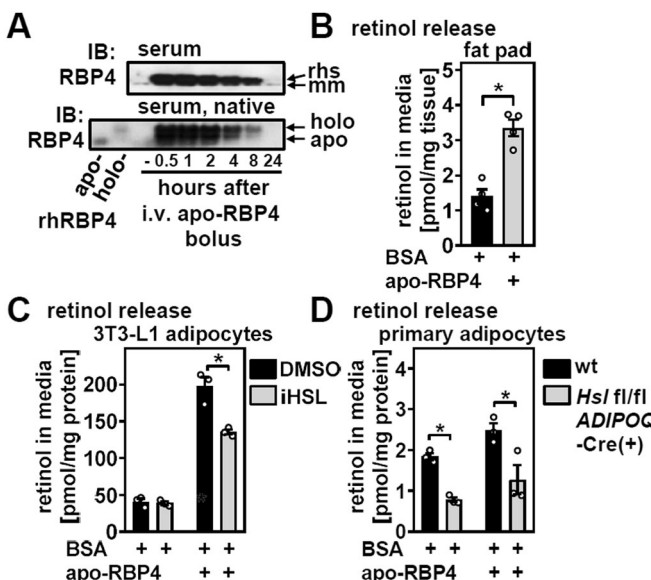

**Figure 6. Retinol-free apo-RBP4 induces HSL-dependent retinol release from adipocytes.**

(A) Serum RBP4 was analyzed by immunoblotting (top) or native PAGE (bottom) at indicated times after i.v. injection of 100 µg of recombinant human RBP4 (rhsRBP4) into the mouse tail vein. Apo- and holo-rhsRBP4 were used as controls for native PAGE. (B) Murine inguinal fat pads were incubated with 4% BSA and 2 µM apo-RBP4 for 2 h as indicated and retinol content in the supernatant analyzed by HPLC. (C) Differentiated 3T3-L1 or (D) in vitro-differentiated primary adipocytes from wild-type (wt) and adipose tissue-specific Hsl knockout mice were loaded with 20 µM retinol overnight. The next day, 3T3-L1 adipocytes were incubated with the HSL inhibitor NNC-0076-0000-0079 and all adipocytes with 4% BSA and 2 µM apo-RBP4 for 2 h as indicated and retinol in the adipocyte supernatant analyzed by HPLC. Data information: In (A), data are representative for two independent experiments. In (B), data show $n = 4$, 4 biological replicates and in (C, D), data show $n = 3$ biological replicates for each condition. Data are represented as individual data points and mean ± s.e.m. and *$P < 0.05$ vs. fat pads/adipocytes incubated with BSA or BSA+apo-RBP4 using an unpaired two-tailed Student's $t$ test (B) or a two-way ANOVA with Sidak's correction for multiple testing (C, D). Source data are available online for this figure.

analyzed whether intravenously (i.v.)-administered recombinant human (rh) apo-RBP4 can yield detectable holo-RBP4 in mouse serum. This was technically possible because of a higher affinity of the used primary antibody to human RBP4 and a slightly different migration behavior of human RBP4 in native PAGE compared to the murine protein. We found that injected apo-RBP4 is easily detectable in mouse serum, circulating for more than 8 h before complete excretion/metabolism 24 h after injection (Fig. 6A, top panel). Non-denaturing PAGE, readily differentiating between holo-and apo-RBP4, demonstrated that the majority of the injected apo-RBP4 circulates as holo-RBP4 (Fig. 6A, bottom panel), in accordance with the notion that apo-RBP4 in the circulation can quickly acquire retinol. Retinol for re-loading may originate from tissues with retinyl ester stores and sufficient REH activity such as WAT, although retinol bound to other binding proteins, due to mass action of the apo-RBP4 bolus, may also contribute.

In order to test whether adipose tissue per se can provide retinol for apo-RBP4 loading, we incubated isolated fat pads with media containing 4% of BSA, a lower-affinity binding protein for retinol (N'Soukpoe-Kossi et al, 2007), with or without 2 µM of apo-RBP4.

Retinol content in fat pad supernatant substantially increased in the presence of apo-RBP4 (Fig. 6B), suggesting that apo-RBP4 triggers retinol release from adipocytes. We then loaded differentiated 3T3-L1 adipocytes (Green and Meuth, 1974) with 20 µM retinol overnight to increase their retinyl ester content, and incubated cells with media containing 4% BSA and apo-RBP4 as indicated. Like in fat pads, apo-RBP4 increased retinol concentrations in the adipocyte culture media, and this elevation was reduced by inhibiting HSL (Fig. 6C). Similar effects were observed in in vitro-differentiated primary adipocytes derived from wt or adipocyte-specific Hsl knockout mice that were preloaded with retinol overnight, where retinol concentrations in adipocyte culture media were increased in the presence of apo-RBP4 but lower in the absence of HSL (Fig. 6D). Thus, extracellular apo-RBP4 induces HSL-dependent retinol release from adipocytes.

## HSL, specifically in adipose tissue, is required for maintaining circulating RBP4 and retinol concentrations during fasting

Having established the role of adipocyte-expressed HSL for retinol release in vitro, we next hypothesized that also adipocyte-specific Hsl deletion reduces circulating RBP4 and retinol concentrations during fasting, similar to the effects observed upon its global deletion. Indeed, adiponectin (ADIPOQ) promoter-driven Cre expression (Pajed et al, 2021), deleting HSL in adipose tissue but not liver of mice with floxed Hsl alleles (Fig. EV5A), lowered circulating retinol and RBP4 only in fasted-, but not in ad libitum-fed mice (Fig. 7A–D, RBP4 blots for ad libitum-fed mice shown in Appendix Fig. S3B). Irrespective of the feeding status, mRNA expression and protein abundance of RBP4 in the liver was not affected by adipocyte-specific HSL deletion (Fig. EV5B–E). Similar to mice with global deletion of HSL, mRNA and protein expression of RBP4 and that of canonical PPAR target genes were reduced in pgWAT of fasted Cre(+) mice (Fig. EV5F–I). We conclude that HSL expression in adipocytes is required to maintain circulating retinol and RBP4 concentrations, specifically during fasting, most likely by driving retinol mobilization from adipose retinyl ester stores.

## Adipose tissue-specific deletion of HSL affects retinoid homeostasis in the eye and kidney during fasting

We next addressed whether reduced retinol concentrations in the circulation of fasted mice with adipose tissue-specific deletion of HSL affects other organs. Retinoid content of eyes in Cre(+) mice was unaltered (Fig. 7E), but gene expression of lecithin:retinol acyltransferase (Lrat), a RAR target gene and component of the retinoid visual cycle (Batten et al, 2004), was elevated (Fig. 7F), suggesting an adaptive increase in retinyl ester synthesis to compensate for lower retinol levels in the circulation. In the kidney, with intrinsically lower retinyl ester stores than the eye, retinol concentration was reduced in Cre(+) mice, and retinyl esters showed a similar trend (Fig. 7G, left panel). Since kidney weights of Cre(+) mice were lower (Fig. 7H), consequently when calculated per organ, both the retinol and retinyl ester kidney contents were reduced (Fig. 7G, right panel). Surprisingly, renal gene expression analyses demonstrated a rather consistent upregulation of RAR target genes (Fig. 7I), implying that a mobilization of renal retinoids to stabilize serum retinol

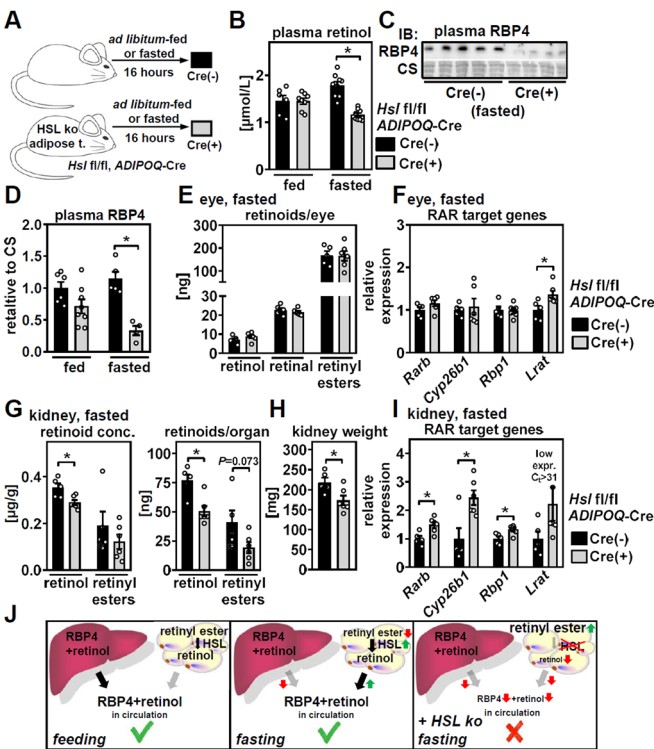

**Figure 7. Adipose tissue-specific HSL deletion lowers circulating retinol and RBP4 and modulates retinoid content and all-*trans* retinoic acid (atRA)-responsive gene expression in the kidney during fasting.**

(A) *ADIPOQ*-Cre(-) and Cre(+) mice with floxed *Hsl* alleles were treated as depicted and (B) plasma retinol analyzed by HPLC. (C) Plasma RBP4 in 16 h-fasted *ADIPOQ*-Cre(−) and Cre(+) mice was determined by immuno-blotting. Coomassie staining (CS) served as loading control. (D) Densitometric analysis of plasma RBP4 in ad libitum-fed and 16 h-fasted mice. *ADIPOQ*-Cre(−) and Cre(+) mice were fasted for 16 h and retinoid content (E) of the eye and (G) of the kidney determined by HPLC and gene expression (F) of the eye and (I) of the kidney analyzed by qPCR. (H) Kidney weights of 16 h-fasted *ADIPOQ*-Cre(−) and Cre(+) mice were determined. Data information: Data in (B, D–I) are represented as individual data points of $n = 6, 8, 9, 9$ (B), $n = 5, 4$ (C), $n = 6, 8, 5, 4$ (D), and $n = 5, 6$ (E–I) of biological replicates and mean ± s.e.m. and *$P < 0.05$ vs. Cre(−) mice using a two-way ANOVA with Sidak's correction for multiple testing (B, D) or an unpaired two-tailed Student's $t$ test (F–I). Source data are available for this figure. (J) Summary, simplified for better illustration: in the fed state, the liver secretes holo-RBP4. In the fasted state, hepatic holo-RBP4 secretion is reduced, whereas HSL in adipose tissue is activated, providing retinol for release and thereby depleting adipose tissue retinyl ester stores. Subjecting mice that lack HSL in adipose tissue to fasting impairs retinyl ester mobilization, reduces retinol release, and lowers circulating retinol and RBP4 which, in turn, modulates retinoid-sensitive gene expression and retinoid content in other organs. Source data are available online for this figure.

concentrations or a decrease in renal retinol uptake associates with an induction of atRA-responsive gene expression. Thus, reduced retinol concentrations in mice with adipose tissue-specific deletion of HSL upon fasting affects retinoid-responsive gene expression and tissue retinoid content of other organs.

## Discussion

We discovered an unrecognized role of adipose tissue retinoid stores in the maintenance of serum retinol and RBP4 levels.

Although HSL was previously linked to retinyl ester hydrolysis in adipocytes (Wei et al, 1997) and adipose tissue (Hansson et al, 2006; Strom et al, 2009), its importance for circulating retinol and RBP4 in fasted mice has not been reported. Selectively during fasting, mice with whole body- or adipocyte-specific deletion of HSL failed to maintain circulating retinol and RBP4 at steady-state levels and, as a consequence, retinoid homeostasis in eye and kidney.

The most likely interpretation of our findings is that upon fasting, when adipocyte HSL activity is intrinsically high due to hormonal stimulation, adipose tissue releases retinol in a HSL-dependent manner into the circulation. As an acceptor, retinol may bind the small but detectable fraction of retinol-free apo-RBP4 in blood. If this release of retinol is hindered, as observed in mice lacking HSL, serum retinol levels decline and apo-RBP4, since not reacquiring retinol, is lost by renal filtration (Goodman, 1980). Quantitatively, retinoid content (retinol + retinyl esters) of all adipose tissue (assuming a ~15% fat mass) is estimated to exceed that of retinol in the circulation by 10-fold, suggesting that an insufficient release upon fasting could indeed explain the observed drop in circulating retinol concentrations. Moreover, HSL-dependent retinol release, thus preventing the loss of RBP4 by renal clearance, also argues for complex re-formation between holo-RBP4 and TTR in the circulation (Quadro et al, 2002; Wei et al, 1995), besides its intracellular assembly for holo-RBP4/TTR secretion by hepatocytes (Bellovino et al, 1996; Melhus et al, 1991). Notably, HSL expression is highest in adipose tissue, suggesting that the mass action "pull" of circulating apo-RBP4 to reacquire retinol in fasted mice may match specifically with increased REH activity and retinol release of adipose tissue, but not with that of other tissues that store retinyl esters. This would explain why an adipocyte-specific HSL knockout leads to similarly reduced retinol and RBP4 in the circulation as observed for its global deletion. The scenario of elevated retinol release from adipose tissue upon fasting is also compatible with the findings that fasting is associated with lower REH activity, reduced TTR expression, and an accumulation of RBP4 protein in the liver, suggesting that hepatic retinol mobilization via RBP4 secretion is reduced upon food withdrawal. Moreover, adipose- but not hepatic retinyl ester stores declined after 24 h of fasting. Thus, fasting appears to induce a relative shift of retinol mobilization from liver to a HSL-dependent mobilization from adipose tissue, warranting rather constant serum levels of retinol and RBP4 and as summarized in Fig. 7J.

An alternative interpretation could be that loss of HSL, either globally or adipocyte-specific, feeds back to reduce hepatic holo-RBP4 secretion specifically during fasting. This could be signaled by adipose tissue-secreted proteins/adipokines or metabolites for interorgan crosstalk. Although we cannot rule out this hepatic contribution, and liver RBP4 appeared indeed reduced in fasted mice globally lacking HSL, adipocyte-specific HSL knockout had no major effect on hepatic expression and abundance of RBP4. Different mechanistic scenarios of altered hepatic RBP4 expression, stability, or secretion could translate into lower RBP4 levels in the circulation during fasting. Nevertheless, since we observed that global HSL deletion did not affect the known induction of hepatic retinyl ester mobilization by feeding a VAD, this suggests that RBP4-mediated retinol mobilization from liver into the circulation is not per se impaired by the loss of HSL.

From a physiological perspective, fasting-induced retinol release by adipocytes seems plausible when considering similarities

between acylglycerols and retinyl esters regarding their chemical structure of esters with fatty acids, storage in liver and adipose tissue, and hydrolysis by the very same enzyme HSL for mobilization from adipocytes. In vitro, recombinant rat HSL showed at least as high activity towards retinyl palmitate when compared to a diglyceride substrate (Strom et al, 2009). Hence, increased acylglycerol mobilization from adipose tissue by activated HSL during fasting appears to be directly linked to elevated mobilization of retinyl esters. In contrast to serum NEFA levels, however, which rise upon fasting and stay high for 24 h in mice, we found that serum retinol remains rather stable. Although different fasting periods may induce some alterations in its serum levels, stable steady-state levels likely result of a fine-tuned coordination between liver and adipose tissue to keep serum retinol within the narrow physiological range. Indeed, we show that cAMP signaling in the liver, which is under the tight endocrine control of glucagon and catecholamines (Lelou et al, 2022), lowers RBP4 secretion by hepatocytes. This likely happens in a coordinated fashion as adipose tissue HSL activity rises. Strikingly, and in contrast to hepatocytes, adipocytes were shown to increase retinol release, thereby depleting their retinyl ester stores upon incubation with cAMP-analogues and in further support of a HSL-dependent process (Wei et al, 1997). Other feedback mechanisms may include the expression and secretion of FGF21 by the liver, known to affect adipose tissue turnover and HSL expression and as previously shown to be regulated by hepatic retinol mobilization (Badman et al, 2007; Inagaki et al, 2007; Kliewer and Mangelsdorf, 2019; Steinhoff et al, 2022b).

Increasing mRNA expression of *Rbp4* in liver and hepatocytes while reducing its secretion, as demonstrated for fasting-induced cAMP signaling, appears counterintuitive. However, intracellular functions of RBP4 for binding/buffering retinol but potentially also fatty acids (Cioffi et al, 2019; Perduca et al, 2018) in liver during fasting, or an increased requirement for rapid RBP4 secretion upon re-feeding may consolidate these observations. Also, the exact mechanisms how adipocytes release retinol, predicted to resemble export mechanisms for other hydrophobic species and presumably independent of STRA6 (Zemany et al, 2014), remain to be determined. Whether adipocyte-expressed RBP4, although not reaching the systemic circulation (Thompson et al, 2017), participates in retinol release is unknown. RBP4 expression in adipose tissue was reduced upon the deletion of HSL, which may be functionally linked to impaired retinol release. On the other hand, our in vitro data show that retinol release by adipocytes clearly depends on HSL, whereas the role of a (secondary) reduction in RBP4 levels in adipocytes remains uncertain. Because 3T3-L1 adipocytes express only marginal amounts of RBP4 (Rosell et al, 2012), the demonstrated impaired retinol release of these adipocytes due to pharmacological inhibition of HSL is likely independent of potential effects on RBP4 abundance.

Adipose tissue-specific deletion of HSL not only reduced retinol and RBP4 in the circulation after an overnight fast, but also affected atRA-responsive gene expression and tissue retinoid content of other organs. It appears plausible that gene expression responds to lower circulating retinol and RBP4 levels more sensitively than the overall tissue retinoid content, and that tissues with higher retinyl ester stores (i.e., the eye) show less disturbances than those with a lower content (i.e., kidney). However, the finding that *Lrat* expression in the eye and most RAR target genes in kidney were

upregulated is surprising. Although one might have hypothesized the opposite, which is reduced RAR target gene expression in a kidney with lower vitamin A content, increased RAR target gene expression, including elevated *Lrat* expression, was previously observed in eyes of mice with severe vitamin A deficiency (Moon et al, 2023). Tissue availability of atRA, the most potent RAR activator, may primarily correlate with flux through retinol mobilization or retinyl ester storage rather than total retinoid abundance. Moreover, we would hypothesize that longer and/or repeated decreases in circulating retinol/RBP4 in these mice, induced by more than a single overnight fast, may trigger additional defects in vitamin A-dependent processes. The lack of a clear retinoid-dependent phenotype in fasted mice with adipose tissue-specific deletion of HSL is a limitation of our study. When triggered, such defects may involve development, the immune system, and, regarding eyesight, a detectable depletion of retinoids in the murine eye with all its functional consequences (Quadro et al, 1999). Exemplary, this was shown for the RBP4 antagonist and RBP4/retinol-lowering compound A1120, where a 12-day treatment was required to decrease retinoids in the murine eyecup by 30–50% (Dobri et al, 2013). Interestingly, mice globally lacking HSL were indeed shown to exhibit visual defects (Yuksel et al, 2023). Although hypothesized to be caused by the deletion of substantial HSL expression in the eye (Yuksel et al, 2023), secondary effects of lower retinol and RBP4 in the circulation during periods of limited food intake may contribute. Additionally, dysregulated retinol mobilization from adipose tissue may, at least in part, contribute to the widely reported, yet incompletely understood observation of elevated RBP4 in the circulation of obese and insulin-resistant mice and humans (Nono Nankam and Bluher, 2021; Yang et al, 2005). De-repressed HSL activity in WAT and retinol release may chronically increase RBP4 saturation and lower apo-RBP4 excretion, thereby contributing to the previously observed accumulation of holo-RBP4 in the circulation of HFD-fed mice (Muenzner et al, 2013). Further studies are needed to test these possibilities.

In regard to adipose tissue health, global or adipocyte-specific HSL deletion has been shown to cause WAT dysfunction that can progress to lipoatrophy in later life (Harada et al, 2003; Osuga et al, 2000; Pajed et al, 2021). This adipose tissue dysfunction coincides with reduced gene expression of a whole variety of lipid metabolic pathways, suggesting that adequate retinol mobilization in adipose tissue is required for local transcriptional responses (Harada et al, 2003; Pajed et al, 2021; Strom et al, 2009), presumably by providing precursors for RAR or RXR activating retinoids. Indeed, transcriptional defects and adipose tissue dysfunction in mice lacking HSL were partially rescued by the administration of atRA (Strom et al, 2009). In humans, a *HSL* null mutation causes dyslipidemia, hepatic steatosis, systemic insulin resistance, and type 2 diabetes. Whether impaired retinyl ester hydrolysis in adipose tissue contributes to these pathologies is unknown and requires further analyses.

In summary, we provide evidence for tissue-specific alterations of retinoid homeostasis and the unequivocal requirement of adipose-tissue HSL for maintaining retinol and RBP4 levels in the circulation of fasted mice. However, insights into retinol flux between liver, adipose tissue, and the circulation, although technically challenging, could further corroborate a shift in retinol mobilization. Nevertheless, the presented findings have important

implications for the understanding of interorgan crosstalk to warrant retinoid homeostasis and widen our perspective on HSL-dependent pathologies.

## Methods

### Animal experiments and characterization

Animal procedures were approved by the corresponding authorities in Berlin/Germany (G 0100-11, G 0130-17, G 0076-22, O 0400-1) and Graz/Austria (39/9/75 ex 2017/18). All mice were on a C57BL/6J genetic background. Mice were housed under standard 12/12 h light/dark cycles, and fed normal chow diet (R/M-Haltung, Ssniff Spezialdiaeten, Soest, Germany, 9% kcal/fat, 25 IU/g vitamin A, before gamma irradiation). Fasting was initiated at indicated Zeitgeber time (ZT) by transferring mice to a new cage with indigestible bedding material and no chow. Mice with global or adipose-tissue-specific deletion of *Hsl* (Pajed et al, 2021; Strom et al, 2009) and respective control mice were housed under 14/10 h light/dark cycles and fed a standard laboratory chow diet (R/M-H Extrudate, V1126-027, 12% kcal/fat, 25 IU/g vitamin A, Ssniff Spezialdiaeten) or a vitamin A-deficient diet as indicated (10 -mm pellets E15311-14, containing <0.12 IU/g vitamin A, Ssniff Spezialdiaeten). Blood glucose was determined by tail vein punctures and glucometer test stripes (Contour, Bayer, Germany). Serum non-esterified fatty acids (NEFA) were analyzed biochemically using the NEFA-HR kit (FUJIFILM Wako Chemicals Europe) and liver triglycerides as described previously (Heidenreich et al, 2017).

### Isolation, culture, and treatment of in vitro-differentiated primary adipocytes, primary hepatocytes, and 3T3-L1 cells

3T3-L1 cells (ATCC, CL-173, (Green and Meuth, 1974)) and a primary stromal vascular fraction (SVF) from inguinal white adipose tissue (ingWAT) of male mice were cultured and differentiated as described previously (Schreiber et al, 2015; Tolkachov et al, 2018; Witte et al, 2015). For ex vivo retinol release, perigonadal white adipose tissue (pgWAT) from male mice was isolated and cut into small pieces of ~3 mm diameter and incubated in DMEM containing low glucose (1 g/L) before 4% fatty acid (FA)-free bovine serum albumin (BSA) and 2 µM apo-RBP4 were added as indicated in the respective figure. Primary hepatocytes were isolated as described elsewhere (Wilson et al, 2010) from male mice. After isolation, hepatocytes attached to the cell culture plate while cultured in DMEM with 4.5 g/L glucose and 10% fetal bovine serum for 5 h. Hepatocytes were then switched to glucose- and serum-free DMEM with 2 mM of sodium pyruvate and 20 mM of sodium lactate as fasting-mimicking conditions. The proteasome inhibitor MG132 (Merck) was used at 10 µM. Depletion of Foxo1 by siRNA (Appendix Table S1) in primary hepatocytes was performed as described before (Heidenreich et al, 2017). Primary hepatocytes and adipocytes were incubated with compounds (all Merck) or apo-RBP4 for indicated time points and concentrations.

### Quantification of retinoids

Serum retinol, retinol, and retinyl esters in tissues of ad libitum-fed and fasted wild-type (wt) mice on normal chow diet were determined by HPLC-PDA as described previously (Muenzner et al, 2013). The retinyl esters quantified were retinyl oleate, -palmitate, and -stearate, accounting together for more than 95% of all retinyl ester stores in liver (D'Ambrosio et al, 2011). Retinoids of plasma and tissue samples from *Hsl* knockout and *Hsl* flox/flox *ADIPOQ*-Cre mice as well as supernatants of 3T3-L1 cells, primary differentiated SVF cells, and fat pad explants were analyzed by HPLC-FD as described previously (Pajed et al, 2019; Wagner et al, 2022). In brief, tissues (20–50 mg) were homogenized in 200 µl PBS, 200 µl ethanol (containing 2.8 µM retinyl acetate as internal standard), and 1 ml n-hexane (containing 1 mM butylhydroxytoluene) using a ball mill (Retsch GmbH, Germany). Phase separation was achieved by centrifugation at $5000 \times g$ at 4 °C for 10 min and the upper organic phase was collected. For repeated extraction, 1 ml n-hexane was added to the remaining tissue homogenate, and vigorously vortexed for 20 s. For phase separation, samples were centrifuged at $5000 \times g$ at 4 °C for 10 min. Organic phases were combined and dried in a speed-vac (Labconco, Kansas City, MO). Plasma (20 µl) and cell supernatants (200–300 µl) were directly n-hexane extracted by vortexing. For retinoid analyses by HPLC-FD, retinoids were separated on a YMC-Pro C18 column (150 × 4.6 mm, S–3 µl, 12 nm, YMC Europe GmbH, Dinslaken, Germany) using a gradient solvent system (flow, 2 ml/min; gradient, 0–2 min 100% methanol, 2–4.2 min 60%/40% methanol/toluene, and 4.2–6 min 100% methanol). Fluorescence was detected at excitation 325 nm/emission 490 nm. The HPLC consisted of a Waters e2695 separation module, a column oven (at 25 °C), and a Waters 2475 fluorescence detector (Waters Corp., Milford, MA). Data were analyzed using Empower 3 chromatography data software (Waters Corp.). Area under the peak was standardized against known amount of the internal standard retinyl acetate. Retinoid content was normalized to mg tissue, whole tissue weight, or cell protein as indicated.

### Determination of retinyl ester hydrolase activity

Measurement of ex vivo retinyl ester hydrolase (REH) activity was performed as previously described (Pajed et al, 2019). In brief, tissue lysates of the liver ($1000 \times g$, supernatant) or WAT ($20,000 \times g$, interphase) were incubated with retinyl palmitate (150 µM) emulsified with phosphatidylcholine (150 µM) in potassium phosphate buffer (pH 7.5, 50 mM) for 1 h at 37 °C. Released retinol was n-hexane extracted and measured by HPLC-FD analysis as described above. Where indicated, the incubation mixtures contained either DMSO (vehicle control), Atglistatin (iATGL; 20 µM), or the HSL inhibitor NNC-0076-0000-0079 (iHSL; 20 µM).

### RNA isolation, cDNA synthesis, and analysis of gene expression by quantitative PCR (qPCR)

Total RNA was purified by spin column kits (VWR), translated to cDNA using MMLV reverse transcriptase (Promega), and analyzed by qPCR as described previously using standard curves for each primer pair and *Rplp0* expression for normalization (Heidenreich et al, 2020). Isolation of RNA, DNAse digestion, cDNA synthesis, and qPCR from liver samples of *Hsl* flox/flox *ADIPOQ*-Cre mice were performed as previously described (Jaeger et al, 2015). Target gene expression was calculated by the $^{\Delta\Delta}$CT method. Expression of

ribosomal housekeeping gene *CycloB* was used for normalization. Primer sequences are listed in Appendix Table S2.

## Protein isolation and immunoblotting

Protein was isolated by standard methods using RIPA buffer. Tissue/cell protein or mouse serum (diluted 1:25) was separated by SDS-PAGE and blotted to a PVDF membrane. For immunoblots analyzing the abundance of denatured TTR monomers, liver/serum samples were incubated in boiling water for 20 min prior loading. Non-denatured PAGE of mouse serum for the detection of holo- and apo-RBP4 was carried out in the absence of SDS and as previously described (Muenzner et al, 2013). Tissue lysates (10–20 µg) of *Hsl* knockout and *Hsl* flox/flox ADIPOQ-Cre mice were prepared as previously described (Pajed et al, 2019). Proteins were separated by 12.5% SDS-PAGE and transferred onto a PVDF membrane. The membrane was blocked with 4–10% non-fat dry milk (dissolved in TST) and incubated with primary antibodies. After incubation with primary antibodies (Appendix Table S3), secondary horseradish peroxidase-coupled antibodies were added, and chemiluminescence was detected (Thermo Scientific). Densitometric analyses were performed by ImageJ (Schneider et al, 2012).

## Synthesis of recombinant apo-RBP4

Recombinant human apo-RBP4 used for the injection into mice was synthesized in *E. coli* as described elsewhere (Kawaguchi and Sun, 2010; Muenzner et al, 2013). Apo-RBP4 for the incubation of fat pad explants and adipocytes in vitro was generated in *Pichia pastoris* according to a published protocol (Wysocka-Kapcinska et al, 2010).

## Statistics

Sample sizes in mouse experiments and the number of replicates in cell culture experiments were based on previous experience with similar studies but not predetermined. No specific guidelines were followed. Animals were randomized into groups of dietary interventions. During mouse characterization, the operator was blinded regarding the mouse genotype. Reported replicates for in vitro cell culture experiments are replicates of independent wells of cultured and treated cells. Data are displayed as individual data points and as mean ± s.e.m. Significance was determined by a two-tailed Student's *t* test or by ANOVA with Dunnett's/Sidak's post hoc tests by GraphPad Prism as described in the respective figure legends, and $P < 0.05$ deemed significant (*$P < 0.05$).

## Data availability

No data were deposited in a public database from this study.

The source data of this paper are collected in the following database record: biostudies:S-SCDT-10_1038-S44319-024-00158-x.

## Peer review information

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

## Acknowledgements

The authors thank Dr. JB Findlay and Dr. D Martin (National University of Ireland, Maynooth, Ireland) for providing *P. pastoris* that expresses human RBP4 and valuable comments on its usage. This work was supported by the German Research Foundation (DFG, project numbers 390217139, 415542650, and 490946138 to MS and 502067018 to SJW). Additional funding was received by the Austrian Science Fund (grant-DOIs https://doi.org/10.55776/I3535 and https://doi.org/10.55776/P34899 to AL). NY and CL were supported by Chinese Scholarship Council stipends. KMP was supported by the Sonnenfeld Foundation Berlin. AH was funded by the Federal Ministry of Education and Research (BMBF 01EA1706). For the purpose of open access, the author has applied a CC BY public copyright license to any Author Accepted Manuscript version arising from this submission.

## Author contributions

**Julia S Steinhoff**: Conceptualization; Data curation; Formal analysis; Investigation; Methodology; Writing—original draft. **Carina Wagner**: Conceptualization; Data curation; Formal analysis; Investigation; Methodology; Writing—review and editing. **Henriette E Dähnhardt**: Data curation; Investigation; Methodology; Writing—review and editing. **Kristina Košić**: Data curation; Investigation; Methodology; Writing—review and editing. **Yueming Meng**: Data curation; Investigation; Methodology. **Ulrike Taschler**: Data curation; Investigation; Methodology; Writing—review and editing. **Laura Pajed**: Investigation; Methodology; Writing—review and editing. **Na Yang**: Data curation; Formal analysis. **Sascha Wulff**: Investigation; Methodology; Writing—review and editing. **Marie F Kiefer**: Data curation; Investigation; Methodology; Writing—review and editing. **Konstantin M Petricek**: Data curation; Investigation; Writing—review and editing. **Roberto E Flores**: Data curation; Methodology. **Chen Li**: Data curation; Formal analysis. **Sarah Dittrich**: Data curation; Formal analysis; Methodology; Writing—review and editing. **Manuela Sommerfeld**: Data curation; Formal analysis; Methodology. **Hervé Guillou**: Formal analysis; Investigation; Writing—review and editing. **Andrea Henze**: Data curation; Methodology; Writing—review and editing. **Jens Raila**: Data curation; Formal analysis; Methodology; Writing—review and editing. **Sylvia J Wowro**: Data curation; Funding acquisition; Methodology; Writing—review and editing. **Gabriele Schoiswohl**: Data curation; Methodology; Writing—review and editing. **Achim Lass**: Conceptualization; Supervision; Funding acquisition; Methodology; Project administration; Writing—review and editing. **Michael Schupp**: Conceptualization; Formal analysis; Supervision; Funding acquisition; Methodology; Writing—original draft; Project administration; Writing—review and editing.

Source data underlying figure panels in this paper may have individual authorship assigned. Where available, figure panel/source data authorship is listed in the following database record: biostudies:S-SCDT-10_1038-S44319-024-00158-x.

## Funding

## Disclosure and competing interests statement

The authors declare no competing interests.

# Expanded View Figures

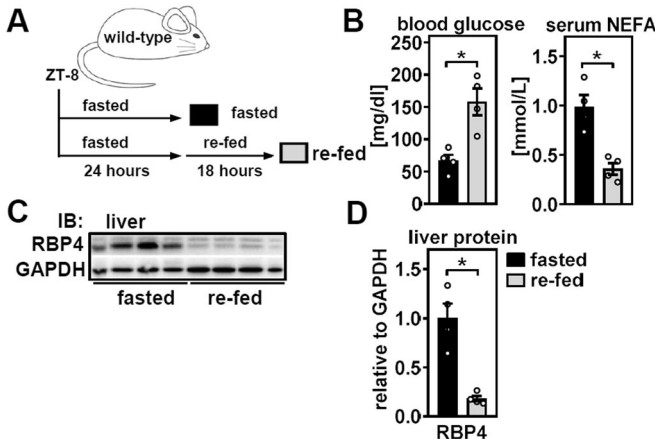

**Figure EV1. Refeeding mice reduces RBP4 protein abundance in liver.**

(A) Mice were fasted or re-fed as depicted. (B) Blood glucose and serum NEFA of fasted and re-fed mice were determined. (C) Hepatic abundance of RBP4 was determined by immunoblotting, GAPDH served as loading control. (D) Densitometric analysis of the blots shown in (C). Data information: Data are represented as individual data points of $n = 4$, 4 (B–D) biological replicates and mean ± s.e.m. and *$P < 0.05$ vs. fasted mice using an unpaired two-tailed Student's $t$ test.

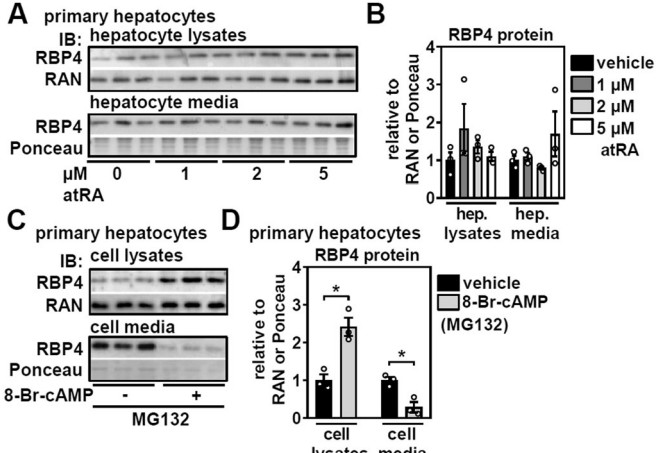

**Figure EV2. RBP4 accumulation and secretion in primary hepatocytes is regulated by cAMP signaling but not by all-*trans* retinoic acid (atRA).**

(A) Primary hepatocytes were incubated with increasing concentrations of atRA and RBP4 protein in cell lysates and media analyzed by immunoblotting, RAN protein and Ponceau membrane staining served as loading controls, respectively. (B) Densitometric analysis of blots shown in (A). (C) Hepatocytes were incubated with 0.5 mM of 8-Br-cAMP for 24 h and RBP4 protein in cell lysates and media analyzed by immunoblotting, RAN protein and Ponceau membrane staining served as loading controls, respectively. 10 µM of the proteasome inhibitor MG132 was added to vehicle and 8-Br-cAMP-treated hepatocytes for the last 4 h before harvesting. (D) Densitometric analysis of blots shown in (C). Data information: Data are represented as individual data points of $n = 3$ for each condition and mean ± s.e.m. and *$P < 0.05$ vs. vehicle treatment using an unpaired two-tailed Student's *t* test (D).

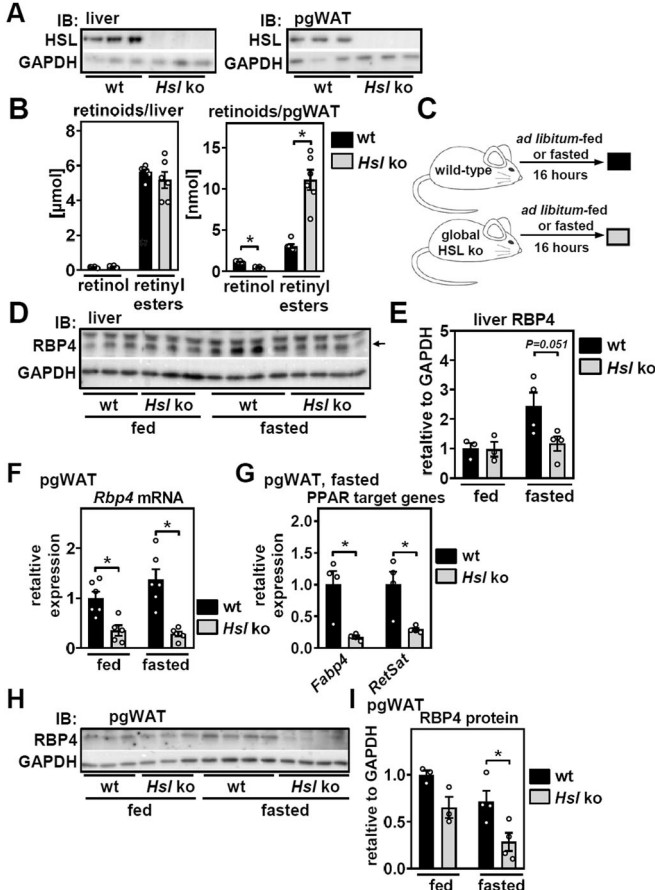

**Figure EV3. Global *Hsl* knockout increases retinyl ester content in WAT but not liver and reduces expression of RBP4 in WAT.**

(A) HSL protein abundance in liver (left panel) and perigonadal white adipose tissue (pgWAT) (right panel) of wild-type (wt) and *Hsl* knockout (ko) mice was determined by immunoblotting. GAPDH protein served as loading control. (B) Tissue retinol and retinyl esters in liver and pgWAT were analyzed by HPLC. Retinoids are shown as n/μmol per total organ. (C) Wt and *Hsl* ko mice were fed and fasted as depicted and (D) abundance of RBP4 protein in livers was determined by immunoblotting. GAPDH served as loading control. (E) Densitometric analysis of blots shown in (D). (F) mRNA expression of *Rbp4* and (G) that of canonical PPAR target genes in pgWAT was determined by qPCR. (H) Hepatic abundance of RBP4 was determined by immunoblotting, GAPDH served as loading control. (I) Densitometric analysis of the blots shown in (H). Data information: Data are represented as individual data points of $n = 3, 3$ (A), $n = 6, 6$ (B), $n = 3, 3, 4, 4$ (D, E), $n = 6, 5, 6, 6$ (F), $n = 4, 4$ (G), and $n = 3, 3, 4, 4$ (H, I) biological replicates and mean ± s.e.m., *$P < 0.05$ vs. wt mice using an unpaired two-tailed Student's *t* test (B, G) or a two-way ANOVA with Sidak's correction for multiple testing (E, F, I).

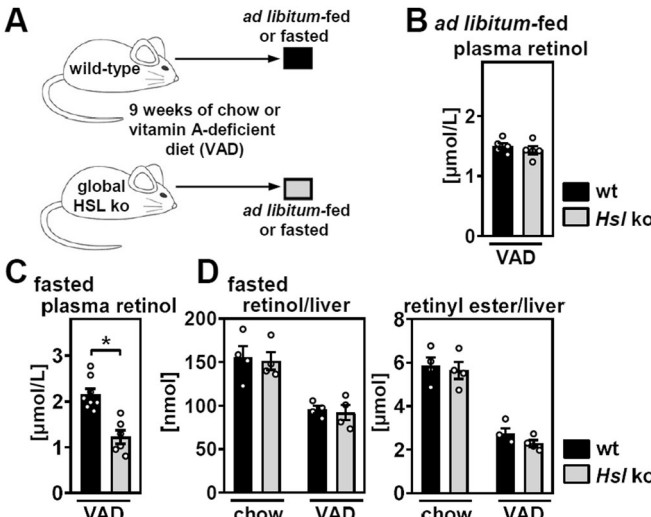

**Figure EV4. Global deletion of HSL does not impair hepatic retinol mobilization upon feeding Vitamin A-deficient diet (VAD).**

(A) Mice of indicated genotype were fed normal chow or VAD and fasted for 16 h or not prior plasma and tissue collection as depicted. (B) Plasma retinol in ad libitum-fed mice on VAD for 9 weeks was determined by HPLC. (C) Plasma retinol in fasted mice on VAD for 9 weeks was determined by HPLC. (D) Retinol (left panel) and retinyl ester content (right panel) of liver after feeding normal chow or VAD in fasted mice was determined by HPLC. Data information: Data are represented as individual data points of $n = 5, 5$ (B), $n = 8, 6$ (C), and $n = 4$ for each group (D) biological replicates and mean ± s.e.m., $*P < 0.05$ vs. wt mice using an unpaired two-tailed Student's $t$ test (C).

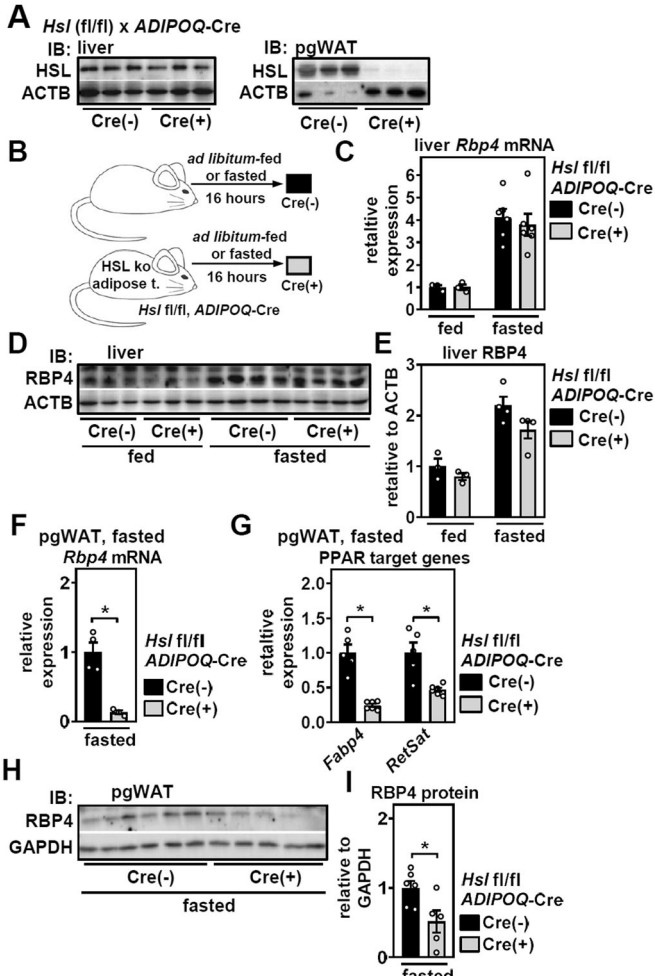

**Figure EV5. Adipose tissue-specific *Hsl* knockout does not affect mRNA and protein expression of hepatic RBP4 but reduces RBP4 levels in WAT.**

(**A**) HSL protein abundance in liver (left panel) and perigonadal white adipose tissue (pgWAT, right panel) of *ADIPOQ*-Cre(−) and Cre(+) mice with floxed *Hsl* alleles was determined by immunoblotting. ACTB protein served as loading control. (**B**) *ADIPOQ*-Cre(-) and Cre(+) mice with floxed *Hsl* alleles were fasted as depicted and (**C**) hepatic mRNA expression of *Rbp4* determined by qPCR. (**D**) Abundance of RBP4 protein in livers of mice described in (**B**) was determined by immunoblotting. ACTB served as loading control. (**E**) Densitometric analysis of blots shown in (**D**). (**F**) mRNA expression of *Rbp4* and (**G**) that of canonical PPAR target genes in pgWAT of fasted mice was determined by qPCR. (**H**) Abundance of RBP4 in pgWAT of fasted mice was determined by immunoblotting, GAPDH served as loading control. (**I**) Densitometric analysis of the blots shown in (**H**). Data information: Data are represented as individual data points of $n = 3, 3, 6, 6$ (**C**), $n = 3, 3, 4, 4$ (**E**), $n = 4, 3$ (**F**), $n = 5, 6$ (**G**), $n = 6, 5$ (**I**) biological replicates and mean ± s.e.m., *$P < 0.05$ vs. Cre(−) mice using an unpaired two-tailed Student's *t* test (**F**, **G**, **I**).

