## [Peer Review File · EMBO Reports]

Adipocyte HSL is required for maintaining circulating vitamin A and RBP4 levels during fasting

Julia Steinhoff, Carina Wagner, Henriette Dähnhardt, Kristina Košić, Yueming Meng, Ulrike Taschler, Laura Pajed, Na Yang, Sascha Wulff, Marie Kiefer, Konstantin Petricek, Roberto Flores, Chen Li, Sarah Dittrich, Manuela Sommerfeld, Herve Guillou, Andrea Henze, Jens Raila, Sylvia Wowro, Gabriele Schoiswohl, Achim Lass, and Michael Schupp

Corresponding author(s): Michael Schupp (michael.schupp@charite.de) , Achim Lass (achim.lass@uni-graz.at)

Review Timeline:

Transfer Date:	27th Mar 24
Editorial Decision:	15th Apr 24
Revision Received:	19th Apr 24
Accepted:	30th Apr 24

Editor: Deniz Senyilmaz Tiebe

Transaction Report: This manuscript was transferred to EMBO reports following peer review at The EMBO Journal.

Note: With the exception of the correction of typographical or spelling errors that could be a source of ambiguity, letters and reports are not edited. Depending on transfer agreements, referee reports obtained elsewhere may or may not be included in this compilation. Referee reports are anonymous unless the Referee chooses to sign their reports.)

Referee #1:

The work by Steinhoff et al. uncovers the role of adipocyte HSL and adipose tissue retinoid stores for retinoid homeostasis during fasting. The authors show that fasting affects retinoid mobilization in a tissue-specific manner. Serum retinol-free apo-RBP4 stimulates retinol release by adipocytes and retinoids are liberated in an HSL-dependent mechanism. This study introduces a novel aspect of biology with the potential to change the way how we think about retinoid metabolism.

Major:

1. In the end, one might wonder how much adipose contributes to total body retinoid metabolism. It is really biologically meaningful? The authors show convincingly that adipose releases retinoids during fasting. As the vast majority is found in the liver and only little compared to that is stored in adipose, it is quite surprising to find that the plasma levels are affected in the HSL KO mice upon fasting. The authors discuss feedback regulation of RBP4 release by the liver and this could also implicate adipokines in the regulation of hepatic retinoid metabolism. An open discussion is warranted here and, ideally, the release of retinoids during fasting by the liver in WT vs adipocyte-HSL KO mice should be experimentally measured to unequivocally settle this issue.
2. Claims made about Figure 7 are only partially supported by the data shown/are hard to retrace (for details see minor points regarding figure 7 below). Altogether, figure 7 is only loosely connected to the first 6. The question of how obesity-linked adipose tissue dysfunction is implicated here should be saved for another day.

Minor:

3. Do HSL KOs or adipocyte-HSL KO mice have any eyesight problems? What would be circumstances under which adipose stores might play a role?
4. Figure 3C - what about dexamethasone alone?
5. In Figure 7D the authors claim a slight decrease in both proteins, RBP4 and TTR, in the liver. While the decrease is visible for RBP4, there is no difference between NC and HFD for TTR, although 7E somehow shows a difference although insignificant.
6. In Figure 7I the authors talk about a visible trend of higher serum retinol in obese, however there is not really a trend, but higher variance.
7. In Figure 7J the authors claim a significant increase in RBP4 and TTR in HFD mice, however although shown in the quantification, the increase is hard to believe as it is not visible on the blot - especially without a loading control.

Referee #2:

The current study identifies a role for adipose tissue retinol/retinyl ester storage in maintaining serum retinol concentration during fasting. Paradoxically, liver RBP4/retinol export appears to

be inhibited, while serum retinol levels remain unchanged during fasting. The authors find a reciprocal reduction in adipose tissue retinol esters by fasting and deletion of Hsl either at the whole body level or only in adipocytes blunts the ability of adipose tissue to replenish serum retinols at the fasted state. In the same vein, the authors demonstrate that apo-RBP4 is able to stimulate retinol release from adipocytes both ex vivo and in vitro.

Adipocytes are known to store a small percentage of retinol in the form of retinyl esters (relative to liver), which can be readily released in vitamin A deficiency (Nutrients 3:27-39). The authors provide solid evidence to suggest that fasting, or reduced dietary vitamin A intake, can trigger a similar mechanism and adipose Hsl is required for retinol release. The study, however, is quite descriptive. Given the fact that it's hard to deplete vitamin A store in mouse studies, the physiological relevance of the observed fasting-induced retinol release by adipose tissue remains unclear. As it stands, the study is more suitable for specialized journals.

Specific comments

1. The entire Figure 3 attempts to address the mechanism through which fasting leads to a 2-fold induction of Rbp4 mRNA expression. However, Figure 1 shows a 4-fold increase in RBP4 protein, likely to due to reduced secretion. This seems to be a more important question, which has not been addressed.
2. Fig. 6 shows that addition of a high concentration of apo-RBP4 promotes retinol release from adipocytes. However, Fig. 2 demonstrates that serum RBP4 levels remain relatively stable while hepatic RBP4 release seems to be inhibited. Does adipose tissue produce sufficient RBP4 to compensate for the difference? If so, how does feeding/fasting affect RBP4 mRNA/protein expression in adipocytes?
3. Similarly, how does adipose tissue Hsl affect serum RBP4 levels?
4. HFD/obesity has been shown to increase serum RBP4, which is believed to contribute to the pathogenesis of obesity-related metabolic dysfunction. What's the physiological relevance of fasting-induced adipose RBP4/retinol release?

Rebuttal letter, EMBOJ-2023-115006

Referee #1

We thank the reviewer for her/his feedback on our manuscript and for acknowledging the novel aspect of biology we are describing and its wider implications for retinoid metabolism. As suggested, we removed the old Figure 7 and focused instead on a more thorough characterization of mice with global or adipose tissue-specific deletion of HSL to substantiate the physiological relevance of our findings. For clarity, we have rearranged the figures for a better separation of results from the different genetic mouse models. Detailed responses to each point raised by the reviewer can be found below.

Major:

1. In the end, one might wonder how much adipose contributes to total body retinoid metabolism. It is really biologically meaningful? The authors show convincingly that adipose releases retinoids during fasting. As the vast majority is found in the liver and only little compared to that is stored in adipose, it is quite surprising to find that the plasma levels are affected in the HSL KO mice upon fasting.

We thank the reviewer for raising this important question. Yes, hepatic retinyl esters represent -in terms of quantity- by far the largest retinoid store. Our analysis of tissue retinoids in *ad libitum*-fed and fasted mice presented in **Figure 4** showed this expected tissue distribution pattern very clearly.

In the circulating blood volume of mice, however, the absolute quantity of retinol can be estimated to be around 0.6-0.9 μg (in a mouse with 25 g of body weight and containing ~ 2 ml of blood (1) with ~ 1 -1.5 $\mu\text{mol/L}$ of retinol). When combining inguinal and perigonadal adipose tissue depots, adipose retinol and retinyl ester content derived from **Figure 4** amounts to a total of ~ 0.2 and ~ 1.6 μg , respectively. It should be noted that this calculation likely underestimates total retinoid content in WAT by quite some margin since A) we only included the parts of both depots that could be excised and weighted, and B) excluded all other murine adipose tissue depots. Assuming a physiological $\sim 15\%$ fat mass of a lean mouse on normal chow and averaging retinoid concentrations of the analyzed depots in **Figure 4**, total adipose tissue retinol and retinyl esters would amount to ~ 1.2 and ~ 9.1 μg , respectively, more than 10-fold the quantity of circulating retinol. In terms of dynamics, we show that during a 24-hour fast, more than 40% of retinol and almost 30% of retinyl esters are mobilized from both depots. Thus, impaired mobilization of these WAT retinoids in mice with global or adipose tissue-specific HSL deficiency could indeed quantitatively explain the observed drop in circulating retinol concentrations upon fasting. Concomitantly, fasting-induced alterations in liver/hepatocytes, including the downregulation of TTR (**Figure 1**) and elevated cAMP signaling (**Figure 3**), likely prevent an increase in hepatic holo-RBP4 secretion to maintain circulating retinol levels despite the more than sufficient abundance of hepatic retinoid stores. Another aspect worth considering are kinetic studies that suggest extensive cycling of retinol between the circulation and tissues (2,3), implying that retinol flux in addition to the total amount present may be biologically relevant. Especially during fasting, loss of adipose HSL may impair retinol cycling between the circulation and adipose tissue by lowering retinol release from retinyl ester stores in WAT. We have now included a short summary of these considerations in the second paragraph of the discussion section on page 15 of the revised manuscript.

In order to test whether impaired mobilization of retinyl esters from WAT, besides lowering plasma retinol (**Figure 5D and 7B**), induces other biological meaningful effects, we exposed a new cohort of mice with or without adipose tissue-specific deletion of HSL to overnight fasting and analyzed eyes and kidneys. Cre(+) showed the expected significantly reduced plasma retinol concentrations (0.60 ± 0.07 vs. 0.90 ± 0.11 $\mu\text{mol/L}$ in Cre(-) mice, $P<0.05$), underlining the robustness of this effect. Fasted Cre(+) mice exhibited no alterations in retinol, retinal, and retinyl esters in the eye compared to their Cre(-) littermates (**new Figure 7E**). However, mRNA expression of the retinyl ester-generating enzyme *Lrat*, a RAR target gene and component of the retinoid visual cycle in the eye, was increased in fasted mice (**new Figure 7F**), indicating that ocular retinoid homeostasis is maintained by adapting *Lrat* expression. In kidney, the concentrations of retinol was reduced and that of retinyl esters trended to be lower (**new Figure 7G**), suggesting that lower plasma retinol concentrations can indeed modulate retinoid content in other tissues. This could be due to reduced delivery, increased catabolism or, as we would deem the most likely explanation, increased mobilization from kidney to prevent a further decrease in plasma retinol. Surprisingly, these changes in retinoids were accompanied by a rather consistent upregulation of RAR target genes (**new Figure 7I**). Although one might have hypothesized the opposite, which is reduced RAR target gene expression in the vitamin A-depleted tissue, increased RAR target gene expression, including elevated *Lrat* expression, was previously observed in eyes of mice with severe vitamin A deficiency (4). Altered gene expression in kidney also provides evidence that the reduction in renal retinol is not simply reflecting lower retinol content of the blood that is perfusing the organ in Cre(+) mice and potentially carried over in the HPLC analysis.

When comparing differences between eye and kidney, however, it appears plausible that we found retinoids reduced in kidney but not the eye since renal retinyl ester stores are much lower than those in the eye, rendering the kidney more sensitive to lower circulating retinol concentrations. In fact, renal retinoid stores may be mobilized more readily into the circulation to maintain circulating retinol whereas the eye, due to its high vitamin A requirements and turnover, functions more likely as a net recipient. Longer and/or repeated decreases in circulating retinol/RBP4 that exceed a single overnight fast may trigger a detectable depletion of retinoids in the murine eye. As an example, this was shown for the RBP4 antagonist and RBP4/retinol-lowering compound A1120. Here, a 12-day treatment with A1120 reduced retinoids in the murine eyecup by 30-50% (5).

Description and discussion of the new data can be found on pages 13 and 17 of the revised manuscript, demonstrating that the observed reduction in circulating retinol does have biological effects in certain tissues.

The authors discuss feedback regulation of RBP4 release by the liver and this could also implicate adipokines in the regulation of hepatic retinoid metabolism. An open discussion is warranted here and, ideally, the release of retinoids during fasting by the liver in WT vs adipocyte-HSL KO mice should be experimentally measured to unequivocally settle this issue.

Although mice with adipose tissue-specific deletion of HSL did not show major alterations in hepatic RBP4 expression/abundance during fasting, we cannot exclude feedbacks from adipose tissue affecting hepatic RBP4 release. We agree with the reviewer that adipose tissue-secreted adipokines may constitute such a feedback. Other feedback signals might include adipose-derived metabolites. HSL-deficient adipose tissue shows reduced lipolysis with a decreased release of non-esterified fatty acids. Reduced adipose tissue lipolysis, in particular during fasting, can attenuate the activation of PPAR α in liver (6). However, PPAR α

appears not to be a regulator of *Rbp4* expression in primary hepatocytes (**Figure 3A**) and liver (**Figure EV5C** and (7)). We discuss these points on pages 15 and 16 of the revised manuscript.

We struggled technically to design a for us feasible experiment to directly assess the release of retinoids during fasting by the liver in WT and adipose tissue-specific HSL ko mice. We therefore tested whether HSL deficiency would impair hepatic retinol mobilization upon another physiologic stimulus, feeding vitamin A-deficient diet (VAD) for extended periods of time (**new Figure EV4A**). VAD for 9 weeks depleted hepatic retinol and retinyl ester by ~40 and ~56%, respectively, when compared to chow-fed mice (**new Figure EV4D**), providing evidence for robust hepatic mobilization to compensate for the lack of nutritional intake. However, mice lacking HSL showed a very similar depletion (**new Figure EV4D**). Also plasma retinol, remaining within the physiologic range, was not affected by the loss of HSL after 9 weeks of VAD in mice fed *ad libitum* (**new Figure EV4B**). These new data show that hepatic retinol mobilization is not *per se* impaired in mice lacking HSL. Also in VAD-fed mice, HSL deletion reduced plasma retinol only in the fasted state (**new Figure EV4C**), highlighting the particular constellation during fasting that triggers the phenotype as delineated above (**Figure 5D-F** and **Figure 7B-D**). These results are described on pages 11 and 12, and discussed on page 16 of the revised manuscript.

2. Claims made about Figure 7 are only partially supported by the data shown/are hard to retrace (for details see minor points regarding figure 7 below). Altogether, figure 7 is only loosely connected to the first 6. The question of how obesity-linked adipose tissue dysfunction is implicated here should be saved for another day.

We agree with the reviewer that the **old Figure 7** is only loosely connected to the rest of the manuscript and removed the figure from the revised manuscript.

Minor:

3. Do HSL KOs or adipocyte-HSL KO mice have any eyesight problems? What would be circumstances under which adipose stores might play a role?

This is in fact an interesting question to be addressed. A very recent study showed that CRISPR/Cas9-generated *Lipe*^{-/-} mice with a frame shift mutation in the *Lipe* gene, leading to a predicted truncated HSL protein, develop an accumulation of subretinal microglia, a retinal degeneration with decreased visual function, and an abnormal retinal lipid profile (8). The authors of this study hypothesized that these defects are most likely due to direct effects of HSL on retinal/RPE lipid homeostasis and retinal health since they demonstrate substantial expression of HSL within these compartment in mice (8). Retinoids in the eye or the systemic circulation were not determined.

However, a global deletion or inactivation of HSL cannot differentiate between direct (eye-autonomous) and indirect effects (e.g. via impaired retinol mobilization in adipose tissue) on eyesight. This would require the analysis of mice with adipose tissue-specific deletion of HSL where its activity in the eye is not ablated. As described in our response to major point 1, Cre(+) mice exhibited no alterations in retinol, retinal, and retinyl esters in the eye compared to their Cre(-) littermates after an overnight fast that lowered circulating retinol (**new Figure 7E**). As described above, the observed upregulation of *Lrat* expression in the eye may counteract a depletion of retinoids. Based on the unaltered abundance of retinoids, we would hypothesize that eyesight of Cre(+) is not compromised after a single overnight fast. As stated above, longer and/or repeated fasting periods or an overall reduced abundance and delivery of vitamin A to the eye due to other genetic/dietary interventions (4) may indeed

cause detectable differences in retinoids of the eye that, as a consequence, impair eyesight of these mice. Regarding wild type mice, pharmacological inhibition of adipose tissue HSL during longer and/or repeated fasting periods may induce similar effects. A short summary of these points can be found in the discussion section on page 17 of the revised manuscript.

4. Figure 3C - what about dexamethasone alone?

We performed the cell experiment also with dexamethasone alone and have added this condition to **revised Figure 3C**. Dexamethasone treatment alone did very little to both *Pck1* and *Rbp4* expression in primary hepatocytes, implying that cAMP signaling is the major effector.

5. In Figure 7D the authors claim a slight decrease in both proteins, RBP4 and TTR, in the liver. While the decrease is visible for RBP4, there is no difference between NC and HFD for TTR, although 7E somehow shows a difference although insignificant.

6. In Figure 7I the authors talk about a visible trend of higher serum retinol in obese, however there is not really a trend, but higher variance.

7. In Figure 7J the authors claim a significant increase in RBP4 and TTR in HFD mice, however although shown in the quantification, the increase is hard to believe as it is not visible on the blot - especially without a loading control.

We thank the reviewer for points 5-7 and commenting critically on the data shown in **old Figure 7**, which we have now removed from the revised manuscript. Since this old Figure 7 is not part of this manuscript version anymore, we do not further elaborate on these points.

Referee #2

We thank the reviewer for her/his valuable comments on our manuscript. We are grateful for the assessment that we provide solid evidence on the underlying mechanisms of fasting-induced retinol release from adipose tissue that requires HSL. As suggested by reviewer 1, we have removed the old Figure 7. In the revision process we have focused on a more in-depth characterization of mice with global or adipose tissue-specific deletion of HSL to substantiate the physiological relevance of our findings. For clarity, we have rearranged the figures for a better separation of results from the different genetic mouse models. Our responses to each point raised by the reviewer are detailed below.

The study, however, is quite descriptive. Given the fact that it's hard to deplete vitamin A store in mouse studies, the physiological relevance of the observed fasting-induced retinol release by adipose tissue remains unclear.

We agree with the reviewer that depleting vitamin A stores in mouse studies is hard to achieve. In part, this is due to the sheer abundance of hepatic retinyl ester stores in mice (**Figure 4**) and the homeostatic regulation that warrants an extremely stable and tightly regulated steady state level of ~0.5-1.5 μM retinol in the circulation (**Figure 1C**). In situations of insufficient ingestion, the general consensus is that circulating retinol drops only after the almost complete exhaustion of hepatic and extra-hepatic retinoid stores (9,10). We would argue that these prerequisites render our findings even more intriguing, since fasting, combined with loss of HSL in adipose tissue, results in a remarkable decrease in circulating retinol and RBP4.

Acknowledging the reviewer's criticism regarding physiological relevance, we exposed a new cohort of mice with or without adipose tissue-specific deletion of HSL to overnight fasting and analyzed eyes and kidneys. Cre(+) showed the expected significantly reduced plasma retinol concentrations (0.60 ± 0.07 vs. 0.90 ± 0.11 $\mu\text{mol/L}$ in Cre(-) mice, $P<0.05$), underlining the robustness of this effect. Fasted Cre(+) mice exhibited no alterations in retinol, retinal, and retinyl esters in the eye compared to their Cre(-) littermates (**new Figure 7E**). However, mRNA expression of the retinyl ester-generating enzyme *Lrat*, a RAR target gene and component of the retinoid visual cycle in the eye, was increased in fasted mice (**new Figure 7F**), indicating that ocular retinoid homeostasis is maintained by adapting *Lrat* expression. In kidney, the concentrations of retinol was reduced and that of retinyl esters trended to be lower (**new Figure 7G**), suggesting that lower plasma retinol concentrations can indeed trigger a decrease in retinoids in other tissues. This could be due to reduced delivery, increased catabolism or, as we would deem the most likely explanation, increased mobilization from kidney to prevent a further decrease in plasma retinol. Surprisingly, these changes in retinoids were accompanied by a rather consistent upregulation of RAR target genes (**new Figure 7I**). Although one might have hypothesized the opposite, which is reduced RAR target gene expression in the vitamin A-depleted tissue, increased RAR target gene expression, including elevated *Lrat* expression, was previously observed in eyes of mice with severe vitamin A deficiency (4). Altered gene expression in kidney also provides evidence that the reduction in renal retinol is not simply reflecting lower retinol content of the blood that is perfusing the organ in Cre(+) mice and carried over in the HPLC analysis.

When comparing differences between eye and kidney, however, it appears plausible that we found retinoids reduced in kidney but not the eye since renal retinyl ester stores are much lower than those in the eye, rendering the kidney more sensitive to lower circulating retinol concentrations. In fact, renal retinoid stores may be mobilized more readily into the circulation to maintain circulating retinol whereas the eye, due to its high vitamin A turnover, functions more likely as a net recipient. Longer and/or repeated decreases in circulating

retinol/RBP4 that exceed a single overnight fast may trigger a detectable depletion of retinoids in the murine eye. As an example, this was shown for the RBP4 antagonist and RBP4/retinol-lowering compound A1120. Here, a 12-day treatment with A1120 reduced retinoids in the murine eyecup by 30-50% (5).

Description and discussion of the new data can be found on pages 13 and 17 of the revised manuscript, demonstrating that the observed reduction in circulating retinol does have biological effects in certain tissues.

Specific comments

1. The entire Figure 3 attempts to address the mechanism through which fasting leads to a 2-fold induction of Rbp4 mRNA expression. However, Figure 1 shows a 4-fold increase in RBP4 protein, likely to due to reduced secretion. This seems to be a more important question, which has not been addressed.

We thank the reviewer for pointing out this discrepancy. Yes, in **Figure 3** we provide evidence that cAMP and FOXO1 signaling in hepatocytes induces *Rbp4* mRNA expression, likely explaining the ~2 fold increase of its mRNA expression in livers of mice fasted for 24 hours (**Figure 1D**). However, **Figure 3E** and its densitometric analysis **3F** show that cAMP also strongly decreases the secretion of RBP4 protein by primary hepatocytes into the media. Furthermore, in these experiments we observed only a trend towards increased cellular RBP4 protein levels (**Figure 3E, F**), not reflecting the robust accumulation of RBP4 protein in fasted liver (**Figure 1F**). Since RBP4 is a highly expressed protein that may be subject to proteasomal degradation in isolated primary hepatocytes, we repeated this experiment in the presence of the proteasome inhibitor MG132 for the last 4 hours before harvesting (**new Figure EV2C, D**). Although the observed reduction in RBP4 secretion upon 8-Br-cAMP was comparable to the one observed before, the presence of MG132 led to a much more pronounced accumulation of RBP4 protein in 8-Br-cAMP-treated hepatocytes (**new Figure EV2C, D**). We conclude that higher levels of RBP4 protein in primary hepatocytes are targeted by proteasomal degradation that, at least in part, masks the effect of 8-Br-cAMP on its cellular accumulation. We describe these new results on page 10 of the revised manuscript. Admittedly, there might be other yet unidentified mechanisms that contribute to the accumulation of RBP4 in liver of fasted mice.

2. Fig. 6 shows that addition of a high concentration of apo-RBP4 promotes retinol release from adipocytes. However, Fig. 2 demonstrates that serum RBP4 levels remain relatively stable while hepatic RBP4 release seems to be inhibited. Does adipose tissue produce sufficient RBP4 to compensate for the difference?

Figure 6 demonstrates that extracellular apo-RBP4 indeed promotes retinol release from adipocytes. Although RBP4 is also expressed by adipocytes, it is likely restricted to intracellular or local actions within adipose tissue since mice with a hepatocyte-specific deletion of RBP4 have no detectable RBP4 in their circulation (11). Thus, adipocyte-expressed RBP4 does not reach the systemic circulation in mice. As we show *in vitro* and *in vivo*, retinol-free apo-RBP4 can acquire retinol, suggesting that the small, but detectable amount of apo-RBP4 (**Figure 2B**) in the circulation of fasted mice 'pulls out' retinol via mass action. How retinol exactly exists the adipocyte and whether it involves specialized proteins is still unknown.

If so, how does feeding/fasting affect RBP4 mRNA/protein expression in adipocytes?

We thank the reviewer for this interesting question. In additional experiments we have analyzed RBP4 mRNA/protein expression in adipose tissue of *ad libitum*-fed and fasted mice with global deletion of HSL. Feeding/fasting had no significant effect on *Rbp4* mRNA expression in perigonadal WAT, whereas HSL deficiency strongly reduced *Rbp4* mRNA levels under both conditions (**new Figure EV3F**). Similar results were obtained for RBP4 protein with a more pronounced and significant reduction during fasting (**new Figure EV3H, I**). We also analyzed the effect of adipose tissue-specific deletion of HSL and, due to limited availability of mice with the required genotype, restricted the analysis to the fasted state. As observed for mice with global deletion, adipose tissue-specific deletion of HSL strongly reduced both mRNA and protein expression of RBP4 in perigonadal WAT (**new Figure EV5F, H, I**).

Mechanistically, reduced lipolysis in the absence of HSL may lower the activation of PPAR γ in WAT since we observed canonical PPAR γ target genes like *Fabp4* and *RetSat* downregulated (**new Figure EV3G, EV5G**) and in accordance with earlier reports (12,13). *Rbp4* expression in primary white adipocytes has previously been shown to be induced by PPAR γ agonists (14).

We are really grateful that the reviewer raised this question and made us perform these analyses. It is tempting to speculate on the function of RBP4 protein in adipocytes and the consequences of its downregulation upon the loss of HSL: RBP4 could act as intracellular binding protein for retinol and its downregulation may correlate with less retinol being liberated from stored retinyl esters due to the deletion of HSL. In a broader perspective, the downregulation of adipose RBP4 may even contribute to impaired retinol release of adipose tissue lacking HSL. Arguing against this possibility are our data shown in **Figure 6C**, where a short-term incubation with an HSL inhibitor (iHSL) reduces apo-RBP4-induced retinol release. Firstly, the used iHSL exposure time of 2 hours is probably too short to induce a major drop in RBP4 protein in 3T3-L1 adipocytes and secondly, these cells are known for rather low RBP4 expression levels (14). However, RBP4 downregulation might contribute to impaired retinol release of HSL-deficient primary adipocytes and adipose tissue *in vivo*. Furthermore and regarding the notion of RBP4 as an intracellular binding protein for retinol, several specialized cellular retinol binding proteins are expressed in adipocytes that are thought to control retinol trafficking within the cell instead of RBP4 (15,16). There are also reports on retinol-independent functions of RBP4 as a binding protein for certain fatty acids (17-20), and RBP4 was shown to stimulate basal lipolysis of human adipocytes (21). Taken together one could conclude that there is a strong link between (HSL-mediated) lipolytic activity, acylglycerol and retinyl ester hydrolysis, and RBP4 expression in adipocytes but more studies are needed to functionally link them. We summarize these findings briefly in the first paragraph on page 17 of the revised manuscript.

3. Similarly, how does adipose tissue Hsl affect serum RBP4 levels?

We show that deletion of HSL globally or specifically in adipose tissue lowers circulating RBP4 in fasted-, but not in *ad libitum*-fed mice (**Figure 5E, 5F and Figure 7C, 7D**), most likely by decreasing retinol release of WAT that renders circulating retinol-free apo-RBP4 susceptible to renal filtration/excretion, as discussed on page 16 of the revised manuscript.

If the reviewer refers to the effect of adipose tissue Hsl on WAT RBP4 expression, we describe and discuss these new data for point 2.

4. HFD/obesity has been shown to increase serum RBP4, which is believed to contribute to the pathogenesis of obesity-related metabolic dysfunction. What's the physiological relevance of fasting-induced adipose RBP4/retinol release?

This is certainly an interesting question. Our observation that mice lacking HSL expression in adipocytes are not able to maintain circulation retinol levels points towards an essential role of adipose tissue for the maintenance of whole body retinoid homeostasis. In addition to decreased circulating retinol and RBP4 levels in these mice, in experiments performed during the revision process we observed decreased renal retinoids, and changes in the expression of several RAR target genes in the kidney as well as of *Lrat* expression in the eye. Thus, an impairment of fasting-induced adipose retinol release, besides lowering circulating retinol concentrations, does show relevant effects also in other organs.

As discussed above, longer and/or repeated decreases in circulating retinol/RBP4 that exceed a single overnight fast may trigger even more defects in vitamin A-dependent processes including eyesight, development, and the immune system.

Since we followed the suggestion of reviewer 1 to remove the rather loosely connected **old Figure 7**, we will only briefly comment on the relevance of fasting-induced adipose tissue retinol release for obesity-related metabolic dysfunction: Obesity-induced insulin resistance is known to modulate HSL activity in WAT (22-24) and accompanied by enlarged adipocytes and increased lipolysis that promote elevated circulating triglycerides, non-esterified fatty acids, and glycerol. Consequently, this may be accompanied by increased retinol release from adipocytes, thereby leading to increased circulating RBP4 protein levels over disease progression. This mechanistic scenario may explain the underlying cause and we have included a brief discussion of this matter on pages 17 and 18 of the revised manuscript. However, further studies are required to experimentally dissect the links between fasting-induced adipose tissue retinol release and obesity-related metabolic dysfunction.

References

1. How much blood can I take from a mouse without endangering its health? *The Jackson Laboratory, NEWS, News and Insights* (2005) <https://www.jax.org/news-and-insights/2005/october/how-much-blood-can-i-take-from-a-mouse-without-endangering-its-health>
2. Tan, L., Wray, A. E., Green, M. H., and Ross, A. C. Retinol kinetics in unsupplemented and vitamin A-retinoic acid supplemented neonatal rats: a preliminary model. *J Lipid Res* (2014) 55, 1077-1086
3. Cifelli, C. J., Green, J. B., and Green, M. H. Use of model-based compartmental analysis to study vitamin A kinetics and metabolism. *Vitam Horm* (2007) 75, 161-195
4. Moon, J., Zhou, G., Jankowsky, E., and von Lintig, J. Vitamin A deficiency compromises the barrier function of the retinal pigment epithelium. *PNAS Nexus* (2023) 2, pgad167
5. Dobri, N., Qin, Q., Kong, J., Yamamoto, K., Liu, Z., Moiseyev, G., Ma, J. X., Allikmets, R., Sparrow, J. R., and Petrukhin, K. A1120, a nonretinoid RBP4 antagonist, inhibits formation of cytotoxic bisretinoids in the animal model of enhanced retinal lipofuscinogenesis. *Invest Ophthalmol Vis Sci* (2013) 54, 85-95
6. Fougerat, A., Schoiswohl, G., Polizzi, A., Regnier, M., Wagner, C., Smati, S., Fougerat, T., Lippi, Y., Lasserre, F., Raho, I., Melin, V., Tramunt, B., Metivier, R., Sommer, C., Benhamed, F., Alkhoury, C., Greulich, F., Jouffe, C., Emile, A., Schupp, M., Gourdy, P., Dubot, P., Levade, T., Meynard, D., Ellero-Simatos, S., Gamet-Payraastre, L., Panasyuk, G., Uhlenhaut, H., Amri, E. Z., Cruciani-Guglielmacci, C., Postic, C., Wahli, W., Loiseau, N., Montagner, A., Langin, D., Lass, A., and Guillou, H. ATGL-dependent white adipose tissue lipolysis controls hepatocyte PPAR α activity. *Cell Rep* (2022) 39, 110910
7. Smati, S., Régnier, M., Fougerat, T., Polizzi, A., Fougerat, A., Lasserre, F., Lukowicz, C., Tramunt, B., Guillaume, M., Burnol, A. F., Postic, C., Wahli, W., Montagner, A., Gourdy, P., and Guillou, H. Regulation of hepatokine gene expression in response to fasting and feeding: Influence of PPAR- α and insulin-dependent signalling in hepatocytes. *Diabetes Metab* (2020) 46, 129-136
8. Yuksel, S., Aredo, B., Zegeye, Y., Zhao, C. X., Tang, M., Li, X., Hulleman, J. D., Gautron, L., Ludwig, S., Moresco, E. M. Y., Butovich, I. A., Beutler, B. A., and Ufret-Vincenty, R. L. Forward genetic screening using fundus spot scale identifies an essential role for Lipe in murine retinal homeostasis. *Commun Biol* (2023) 6, 533
9. Quadro, L., Hamberger, L., Colantuoni, V., Gottesman, M. E., and Blaner, W. S. Understanding the physiological role of retinol-binding protein in vitamin A metabolism using transgenic and knockout mouse models. *Molecular Aspects of Medicine* (2003) 24, 421-430
10. Quadro, L., Hamberger, L., Gottesman, M. E., Wang, F., Colantuoni, V., Blaner, W. S., and Mendelsohn, C. L. Pathways of vitamin A delivery to the embryo: insights from a new tunable model of embryonic vitamin A deficiency. *Endocrinology* (2005) 146, 4479-4490
11. Thompson, S. J., Sargsyan, A., Lee, S. A., Yuen, J. J., Cai, J., Smalling, R., Ghyselincq, N., Mark, M., Blaner, W. S., and Graham, T. E. Hepatocytes Are the Principal Source of Circulating RBP4 in Mice. *Diabetes* (2017) 66, 58-63
12. Pajed, L., Taschler, U., Tilp, A., Hofer, P., Kotzbeck, P., Kolleritsch, S., Radner, F. P. W., Pototschnig, I., Wagner, C., Schratz, M., Eder, S., Huetter, S., Schreiber, R., Haemmerle, G., Eichmann, T. O., Schweiger, M., Hoefler, G., Kershaw, E. E., Lass, A., and Schoiswohl, G. Advanced lipodystrophy reverses fatty liver in mice lacking adipocyte hormone-sensitive lipase. *Commun Biol* (2021) 4, 323
13. Strom, K., Gundersen, T. E., Hansson, O., Lucas, S., Fernandez, C., Blomhoff, R., and Holm, C. Hormone-sensitive lipase (HSL) is also a retinyl ester hydrolase: evidence from mice lacking HSL. *FASEB J* (2009) 23, 2307-2316
14. Rosell, M., Hondares, E., Iwamoto, S., Gonzalez, F. J., Wabitsch, M., Staels, B., Olmos, Y., Monsalve, M., Giralt, M., Iglesias, R., and Villarroya, F. Peroxisome proliferator-activated receptors- α and - γ , and cAMP-mediated pathways, control retinol-binding protein-4 gene expression in brown adipose tissue. *Endocrinology* (2012) 153, 1162-1173
15. Zizola, C. F., Frey, S. K., Jitngarmkusol, S., Kadereit, B., Yan, N., and Vogel, S. Cellular retinol-binding protein type I (CRBP-I) regulates adipogenesis. *Mol Cell Biol* (2010) 30, 3412-3420
16. Zizola, C. F., Schwartz, G. J., and Vogel, S. Cellular retinol-binding protein type III is a PPAR γ target gene and plays a role in lipid metabolism. *Am J Physiol Endocrinol Metab* (2008) 295, E1358-1368
17. Perduca, M., Nicolis, S., Mannucci, B., Galliano, M., and Monaco, H. L. Human plasma retinol-binding protein (RBP4) is also a fatty acid-binding protein. *Biochim Biophys Acta* (2018) 1863, 458-466
18. Perduca, M., Nicolis, S., Mannucci, B., Galliano, M., and Monaco, H. L. High resolution crystal structure data of human plasma retinol-binding protein (RBP4) bound to retinol and fatty acids. *Data Brief* (2018) 18, 1073-1081
19. Huang, H. J., Nanao, M., Stout, T. J., and J., R. Identification of a Non-Retinoid Compound and Fatty Acids as Ligands for Retinol Binding Protein 4 and Their Implications in Diabetes. (2010)
20. Nanao, M., Mercer, D., Nguyen, L., Buckley, D., and Stout, T. J. Crystal Structure of Rbp4 Bound to Oleic Acid. (2010)
21. Kilcarslan, M., de Weijer, B. A., Simonyté Sjödin, K., Aryal, P., Ter Horst, K. W., Cakir, H., Romijn, J. A., Ackermans, M. T., Janssen, I. M., Berends, F. J., van de Laar, A. W., Houdijk, A. P., Kahn, B. B., and Serlie, M. J. RBP4 increases lipolysis in human adipocytes and is associated with increased lipolysis and hepatic insulin resistance in obese women. *Faseb j* (2020) 34, 6099-6110
22. Recazens, E., Mouisel, E., and Langin, D. Hormone-sensitive lipase: sixty years later. *Prog Lipid Res* (2021) 82, 101084
23. Harada, K., Shen, W. J., Patel, S., Natsu, V., Wang, J., Osuga, J., Ishibashi, S., and Kraemer, F. B. Resistance to high-fat diet-induced obesity and altered expression of adipose-specific genes in HSL-deficient mice. *Am J Physiol Endocrinol Metab* (2003) 285, E1182-1195
24. Park, S. Y., Kim, H. J., Wang, S., Higashimori, T., Dong, J., Kim, Y. J., Cline, G., Li, H., Prentki, M., Shulman, G. I., Mitchell, G. A., and Kim, J. K. Hormone-sensitive lipase knockout mice have increased hepatic insulin sensitivity and are protected from short-term diet-induced insulin resistance in skeletal muscle and heart. *Am J Physiol Endocrinol Metab* (2005) 289, E30-39

Referee #1:

I think the authors have done a great job ironing out the few flaws and logical gaps with this revision. Well done study with new insights into lipid metabolism and Vit A biology.

Referee #2:

The authors have performed additional experiments in response to review comments. The main issue related to the significance of the finding/physiological relevance remains unresolved. Notably, in Fig. 7, reduced fasting serum RBP4 in fat-specific HSLKO mice led to a minimal effect in the eye. Retinol concentration was lower in the kidney. However, the expression of RAR target genes was up-regulated. The authors stated that this was likely a compensatory effect without providing any evidence. As a result, the study remains descriptive. Although the finding is somewhat interesting, the scope of the current study may be more suitable for specialized journals.

Dear Prof. Schupp,

Thank you for submitting your revised manuscript. It has now been seen by two of the original referees.

As you can see, the referees find that the study is significantly improved during revision and recommend publication. However, I need you to address the points below before I can accept the manuscript.

- Please provide a textual response to the remaining concerns of referee #2 and add a discussion point on the caveats pointed out by referee #2.
- We note that the funding information in the Acknowledgements section is incomplete - WO 2624/1-1 is currently missing.
- Please provide 3-5 keywords for your study. These will be visible in the html version of the paper and on PubMed and will help increase the discoverability of your work.
- Please rename the Literature section as References.
- Please remove the Author Contributions section from the manuscript.
- Please rename the Conflicts of Interests section as "Disclosure Statement and Competing Interests".
- We note the phrase 'data not shown' on pages 9, 11, 13 and 16, which is not allowed as per journal policy. Please either show the data or remove the statements.
- Please add page numbers to the Appendix File.
- We note that there are Source Data in the Appendix File, which should be removed and re-submitted as Source Data as one zip file per figure. Please refer to the recent email from our Source Data Coordinator Dr. Hannah Sonntag.
- Our production/data editors have asked you to clarify several points in the figure legends:
 1. Please note that a separate 'Data Information' section is required in the legends of figures 1b, d-e, g; 2a-d; 3a-i; 5a-f; 6a-d; 7b-i; EV 1b, d; EV 2b, d; EV 3b, e-g, i; EV 4b-d; EV 5c, e-g, i.
 2. Please note that the statistical test information for figure 6b is labelled as 6a in the legend. This needs to be rectified.
- The manuscript sections should be in the following order: Title page - Abstract & Keywords - Introduction - Results - Discussion - Methods - Data Availability - Acknowledgments - Disclosure Statement & Competing Interests - References - Figure Legends - Expanded View Figure Legends.
- Papers published in EMBO Reports include a 'synopsis' and 'bullet points' to further enhance discoverability. Both are displayed on the html version of the paper and are freely accessible to all readers. The synopsis includes a short standfirst summarizing the study in 1 or 2 sentences (max 35 words) that summarize the paper and are provided by the authors and streamlined by the handling editor. I would therefore ask you to include your synopsis blurb and 3-5 bullet points listing the key experimental findings.
- In addition, please provide an image for the synopsis. This image should provide a rapid overview of the question addressed in the study but still needs to be kept fairly modest since the image size cannot exceed 550 (width) x 300-600 (height) pixels.

Thank you again for giving us to consider your manuscript for EMBO Reports, I look forward to your minor revision.

Kind regards,

Deniz Senyilmaz Tiebe

--

Deniz Senyilmaz Tiebe, PhD
Editor
EMBO Reports

Responses to comments

- **Please provide a textual response to the remaining concerns of referee #2 and add a discussion point on the caveats pointed out by referee #2.**

Referee #2 remaining concern: 'The main issue related to the significance of the finding/physiological relevance remains unresolved.'

In the manuscript, we demonstrate that body vitamin A homeostasis is dependent on a crosstalk between liver and adipose tissue retinoid stores, which requires functional HSL in adipose tissue. Consequently, when defective, as in mice with adipocyte-specific deletion of Hsl, this leads to lower circulating levels of retinol and RBP4 in fasted mice and provokes alterations in the expression of retinoic acid-responsive genes and retinoid content in other organs like kidney. We would argue that these observed effects by itself demonstrate a significance/physiological relevance of our findings, regarding the role of adipocyte-expressed HSL and the crosstalk between adipose tissue and the liver for systemic retinoid homeostasis. However, we do not show that these alterations in gene expression and retinoid content result in a phenotype of retinoid-dependent organ dysfunction, which is a limitation of our study and a matter of future studies.

We have included a summary of these points on page 14 of the revised manuscript.

Dear Prof. Schupp,

Thank you for submitting your revised manuscript. I have now looked at everything and all is fine. Therefore, I am very pleased to accept your manuscript for publication in EMBO Reports.

Congratulations on a nice work!

Kind regards,

Deniz Senyilmaz Tiebe

--

Deniz Senyilmaz Tiebe, PhD

Editor

EMBO Reports

--
